# R-WoM: Retrieval-augmented World Model For Computer-use Agents

**Kai Mei**
Rutgers University
kai.mei@rutgers.edu

**Jiang Guo,**[*] **Shuaichen Chang, Mingwen Dong**
AWS Agentic AI
{gujiang, cshuaich, mingwd}@amazon.com

**Dongkyu Lee, Xing Niu & Jiarong Jiang**
AWS Agentic AI
{dkleekr, xingniu, jiarongj}@amazon.com

## Abstract

Large Language Models (LLMs) can serve as world models to enhance agent decision-making in digital environments by simulating future states and predicting action outcomes, potentially eliminating costly trial-and-error exploration. However, this capability is fundamentally limited by LLM's tendency to hallucination and their reliance on static training knowledge, which could lead to compounding errors that inhibit long-horizon simulations. To systematically investigate whether LLMs are appropriate for world modeling, we probe two core capabilities of world models – *future state prediction* and *reward estimation* – through three tasks: next-state identification, full-procedure planning alignment, and milestone transition recognition. Our analysis shows that while LLMs effectively capture immediate next states and identify meaningful state transitions, their performance rapidly degrades in full-procedure planning. This highlights LLMs' limitations in reliably modeling environment dynamics over long horizons. To address these limitations, we propose the Retrieval-augmented World Model (R-WoM), which grounds LLM simulations by incorporating factual, up-to-date knowledge retrieved from external tutorials. Experiments show that R-WoM achieves relative improvements of up to 23.4% and 16.3% on the subsets of OSWorld and Webarena compared to baselines, with particular advantage in longer-horizon simulations.

## 1 Introduction

World models have evolved from early symbolic planning systems to sophisticated neural architectures that learn latent representations of environment dynamics. Model-based reinforcement learning (MBRL) approaches, such as Dreamer v1-3 (Hafner et al., 2019; 2020; 2023) and MuZero (Schrittwieser et al., 2020), learn latent world models to "imagine" trajectories before selecting actions. More recently, Large Language Model (LLM)-based world models (Hao et al., 2023; Wang et al., 2024a; Zhang et al., 2024; Ge et al., 2024) have emerged as a new paradigm, leveraging large-scale pretraining to reason about action consequences in realistic digital environments. They show particular promise for long-horizon planning for browser and computer-use agents, where mentally simulating future states can mitigate irreversibility and reduce costly trial-and-error.

However, due to their inherent tendency toward hallucination and reliance on static parametric knowledge, LLMs perform world modeling in a fundamentally ungrounded manner. Some studies explore the grounding of world models to improve video understanding (Ge et al., 2024) or navigation in text-based simulated environments (Zhou et al., 2024). However, there is still a gap of the world modeling of knowledge in complex multi-turn realistic environments (e.g., computer-use). For example, in an OS environment, as illustrated in Figure 1, without proper grounding, computer-use agents struggle to adapt to environment-specific knowledge and often generate procedural steps that seem coherent but are ultimately infeasible to execute.

---

[*]Corresponding Author. Work done when Kai is an intern at AWS Agentic AI.

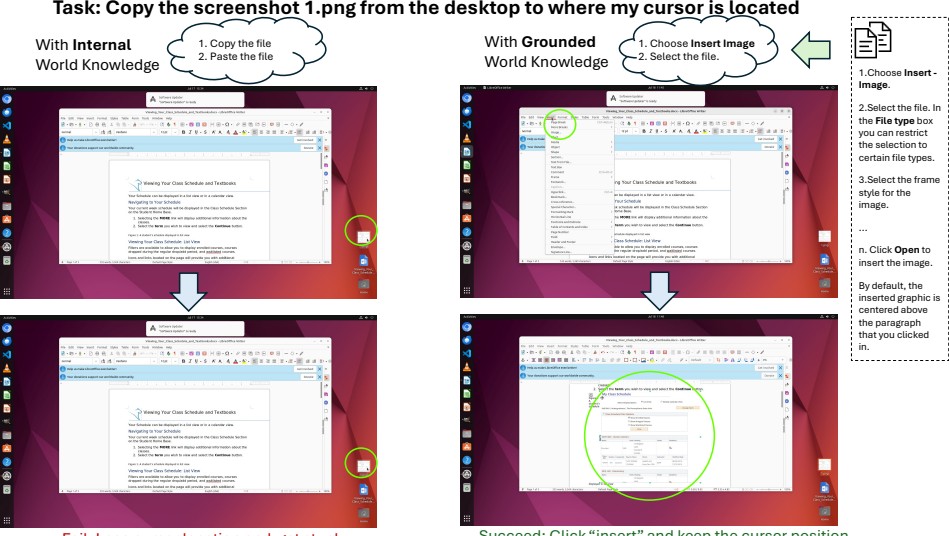

Figure 1: Example task: "Copy the screenshot 1.png from the desktop to where my cursor is located." (**Left:**) Using LLM's internal world knowledge, the agent loses cursor location and gets stuck. (**Right:**) With grounded world knowledge from tutorials, the agent uses the correct "Insert Image" operation while maintaining cursor position.

To systematically investigate whether LLMs can serve as effective world models, we probe two core capabilities: **future state prediction** and **reward estimation**. We design three evaluation tasks: next-state prediction and full-procedure planning alignment to assess LLMs' future state prediction capability; and milestone transition recognition to assess LLMs' reward estimation capability. Our analysis reveals that while LLMs demonstrate strong short-term dynamics understanding, such as identifying state changes and recognizing transition outcomes, they fail to maintain accuracy in full-procedure planning. This performance degradation over longer-horizon reasoning highlights fundamental limitations of LLM-based world modeling.

Motivated by these findings, we propose the **Retrieval-augmented World Model (R-WoM)** framework, which enhances LLM-based simulations by grounding them in external knowledge drawn from environment-specific tutorials. The core insight behind R-WoM is that while LLMs possess broad world knowledge from pretraining, they lack the specific, up-to-date procedural knowledge required for accurate simulation in dynamic digital environments. Recent work suggests that tutorials can function as high-level abstractions of environment dynamics (Xu et al., 2024; Zhang et al., 2025a; Su et al., 2025). However, standard retrieval pipelines often surface noisy or tangential information, which undermines the alignment between retrieved tutorials and the world-modeling process. For instance, a query about "fork chatgpt" might retrieve general Git forking tutorials rather than specific procedures for the current application context. To mitigate this, R-WoM incorporates a *reasoning-based* RAG pipeline that combines query rewriting with LLM-based reranking to improve the relevance of retrieved tutorials. In contrast to prior approaches that rely on computationally expensive iterative rollouts between policy and world models (Gu et al., 2024; Fang et al., 2025), R-WoM leverages the more lightweight long chain-of-thought (CoT) (Guo et al., 2025) reasoning mechanism for multi-step simulation. Moreover, we observe that the use of absolute reward estimation in existing works (Chae et al., 2024; Gu et al., 2024; Fang et al., 2025) could introduce biases and lead to unstable action scoring. To address this limitation, we employ a listwise reward estimation strategy that ranks simulation rollouts relative to each other rather than assigning absolute scores, leading to more robust and consistent action selection. Our key contributions are as follows:

- **Systematic probing of LLMs as world models.** We conduct comprehensive evaluation revealing that while LLMs excel at understanding immediate state changes and local transitions, they critically fail in producing procedures aligned to the environments over long horizons.

- **Retrieval-augmented world modeling framework.** We propose R-WoM, a retrieval-augmented framework that grounds LLM-based world models with external tutorials, enabling environment-specific adaptation through retrieval-augmented simulation and listwise reward estimation.

- **Empirical validation on realistic benchmarks.** We demonstrate R-WoM's effectiveness on two challenging computer-use benchmarks, WebArena (Zhou et al., 2023) and OSWorld (Xie et al., 2024), achieving consistent and substantial improvements (i.e., 5.6% to 23.4%) over competitive baselines, with particular advantages in longer-horizon scenarios.

## 2 BACKGROUND

### 2.1 PROBLEM FORMALIZATION

Given an initial task goal $g$, a computer-use agent interacts with the environment by iteratively receiving observations and executing actions to accomplish the task. Following the notation of prior work (Qin et al., 2025; Fang et al., 2025), we also introduce an intermediate reasoning component thought $t$, to capture thinking process. The resulting interaction trajectory can be expressed as

$$(g, (o_1, t_1, a_1), (o_2, t_2, a_2), \ldots, (o_n, t_n, a_n)), \tag{1}$$

where $o_i$ is the observation at step $i$, $t_i$ is the reasoning thought generated before action selection, and $a_i$ is the executed action. At each step $i$, the LLM-based policy model produces a thought–action pair conditioned on the task goal, the current observation, and the prior interaction history:

$$(t_i, a_i) \sim \pi_p\left(\cdot \mid g, o_i, \{(o_j, t_j, a_j)\}_{j=v}^{i-1}\right), \quad v \in [1, i-1] \tag{2}$$

### 2.2 WORLD MODEL ROLLOUT

In realistic environments, many actions are irreversible or costly to undo, which makes naive trial-and-error exploration infeasible. To address this challenge, researchers explore using a world model (Hafner et al., 2019; 2020; 2023) that can simulate possible futures to be aware of the action outcomes before executing. Formally, at each decision step $i$, given the set of candidate actions along with their thoughts $\mathcal{A}_c = \{(t_i^{(1)}, a_i^{(1)}), (t_i^{(2)}, a_i^{(2)}), \ldots, (t_i^{(m)}, a_i^{(m)})\}$ proposed by policy model $p$ in Equation 2, the world model performs $k$-step lookahead rollouts to estimate the potential outcomes of each action candidate $j \in \{1, 2, \ldots, m\}$:

$$
\begin{aligned}
o_{i+1}^{(j)} &\sim \pi_w(\cdot | g, o_i, t_i^{(j)}, a_i^{(j)}) \\
(t_{i+1}^{(j)}, a_{i+1}^{(j)}) &\sim \pi_w(\cdot | g, o_{i+1}, t_i^{(j)}, a_i^{(j)}) \\
&\vdots \\
o_{i+k}^{(j)} &\sim \pi_w(\cdot | g, o_{i+k-1}^{(j)}, t_{i+k-1}^{(j)}, a_{i+k-1}^{(j)})
\end{aligned} \tag{3}
$$

For each $k$-step rollout trajectory $\hat{\tau}_i^{(j)} = (o_i^{(j)}, t_i^{(j)}, a_i^{(j)}, o_{i+1}^{(j)}, t_{i+1}^{(j)}, a_{i+1}^{(j)}, \ldots, o_{i+k}^{(j)})$, the corresponding rewards are estimated using a model-based (Li et al., 2023; Mahan et al., 2024) or program-based (Lambert et al., 2024; Guo et al., 2025) reward function:

$$r(a^j) = R(\hat{\tau}_i^{(j)}, g) \tag{4}$$

The optimal action is then selected from $\mathcal{A}_c$ based on the highest estimated reward.

$$a_i^* = \arg \max_{(t_i, a_i) \in \mathcal{A}_c} r(a_i) \tag{5}$$

## 3 PRELIMINARY ANALYSIS

We focus on two fundamental capabilities of world models that are critical for computer-use tasks: **future state prediction**, which supports anticipating environment dynamics, and **reward estimation**, which underpins evaluating the outcomes of actions (Hafner et al., 2019; 2020; 2023). Recent work such as WMA (Chae et al., 2024) explores these aspects mainly through next-state identification and immediate reward estimation. However, such analyses do not fully account for the

importance of reasoning across extended horizons. To address this, we design probing tasks tailored to these two capabilities by considering longer planning horizon. Specifically, for future state prediction, we design the task of next-state identification and full-procedure planning alignment, which together capture both short and long horizon dynamics; For reward estimation, we design the task of milestone transition recognition, which assesses models' ability to anticipate the outcomes of intermediate transitions. We apply these probes to three state-of-the-art LLMs, Qwen-2.5-VL-72B (Bai et al., 2025), Claude-3.5-Sonnet[1], and Claude-3.7-Sonnet[2] by sampling trajectories on two challenging browser/computer-use benchmarks: WebArena (Zhou et al., 2023) and OSWorld (Xie et al., 2024). In the following, we introduce these tasks and present the probing analysis, while more details with illustrative examples are provided in Appendix A.1.

## 3.1 NEXT-STATE IDENTIFICATION

To assess the most basic requirement of future state prediction, we follow WMA (Chae et al., 2024) to design this task where models are asked to predict the correct subsequent observation given a current state and action. Given current observation $o_i$ and action $a_i$, the model predicts the correct subsequent observation from two candidates:

$$\hat{o}_{i+1} = \arg \max_{o \in \{o_{i+1}^{\text{true}}, o_{i+1}^{\text{false}}\}} P(o|o_i, t_i, a_i) \tag{6}$$

**Setup:** Given the a $n$-step trajectory, we extract intermediate steps from successful and failed trajectories where $i \in [2, n-2]$ to avoid trivial predictions from initial or terminal states. For each $(o_i, a_i, o_{i+1})$ triplet, we create a negative sample by selecting the most lexically similar observation from the same trajectory. The lexical analysis is conducted using difflib[3], a Python's built-in library. This requires LLMs to distinguish the true next observation $o_{i+1}^{\text{true}}$ from a distractor $o_{i+1}^{\text{false}}$.

**Results:** As shown in Table 1, models achieve relatively strong accuracy overall, i.e., exceeding 75%, indicating they can capture short-term state changes under various lexical similarity levels.

## 3.2 FULL-PROCEDURE PLANNING ALIGNMENT

While next-state identification evaluates whether an LLM can capture immediate state transitions, effective world models must also reason over longer horizons. To probe this ability, we design a plan alignment task, where models are asked to generate execution plans and these plans are evaluated for consistency with realistic environment dynamics. Formally, given a task goal $g$ and an initial observation $o_1$, the model produces an execution plan $\hat{P} = (a_1, a_2, \ldots, a_T)$. The LLM judge then evaluates whether the execution plan conforms to the standard procedure defined in Equation 7.

$$B = \Phi\left(\langle g, o_1 \rangle, \hat{P}, P^*\right) = \begin{cases} \text{True,} & \text{if } \hat{P} \text{ aligns with } P^*, \\ \text{False,} & \text{otherwise.} \end{cases} \tag{7}$$

where $P^*$ denotes the reference procedure derived from environment tutorials. The judgement is based on element attributes (e.g., location, text description, visibility) and operation logic (e.g., feasibility, ordering) with respect to $P^*$.

**Setup:** We sample tasks from WebArena and OSWorld benchmarks. For each task, we manually annotate a reference document chunk that is directly relevant to accomplishing the task under the corresponding environment (e.g., a website or software). More annotation details are in Appendix A.2. Models are then prompted to generate execution plans without access to tutorials, and the generated plans are evaluated by an LLM judge (Claude-3.7-Sonnet by default) for alignment against the reference procedures. More Details of the evaluation prompt are provided in Appendix A.1 and the impact of using other models as the LLM judge are provided in Appendix A.6.

**Results:** Table 1 shows that alignment remains moderate across all models, rarely exceeding 65%. This reveals a clear limitation: while LLMs can list plausible actions, they often fail to maintain

---

[1]https://www.anthropic.com/news/claude-3-5-sonnet
[2]https://www.anthropic.com/news/claude-3-7-sonnet
[3]https://docs.python.org/3/library/difflib.html

Table 1: Probing results across three tasks: next-state identification, full-procedure planning alignment, and milestone transition recognition. All values are percentages.

| Model | Next-state identification (by lexical similarity) | | | | Full-procedure planning alignment | | Milestone transition recognition |
|---|---|---|---|---|---|---|---|
| | [0, 0.8) | [0.8, 0.9) | [0.9, 1] | Overall | w/o retrieval | w/ retrieval | Accuracy |
| Qwen-2.5-VL-72B | 61.1 | 84.8 | 77.6 | 77.0 | 50.0 | 90.0 | 83.7 |
| Claude-3.5-Sonnet | 72.2 | 84.8 | 81.6 | 81.0 | 55.0 | 85.0 | 85.7 |
| Claude-3.7-Sonnet | 88.9 | 87.9 | 83.7 | 86.0 | 65.0 | 95.0 | 86.7 |

procedural coherence or respect environment-specific constraints. After integrating retrieval, however, performance improves across all models, as shown in the w/ retrieval column. The retrieved tutorials provide additional contextual grounding, which helps the model maintain more coherent multi-step reasoning and better align its generated procedures with human-authored references.

### 3.3 MILESTONE TRANSITION RECOGNITION

Aside from probing LLM's capability of capturing future states, we also probe whether models can recognize task-relevant progress, an essential skill for reward estimation in world models. The task evaluates whether models can distinguish promising transition sequences from unproductive ones:

$$\hat{S} = \arg \max_{S \in \{S^{\text{true}}, S^{\text{false}}\}} P(\text{success} \mid S, g) \tag{8}$$

where $S = \{o_i, o_{i+h}, o_{i+2h}, \dots, o_{i+(l-1)h}\}$ denotes a subsequence of length $l$ sampled at interval $h$ from the full trajectory.

**Setup:** We sample sequences of $l = 3$ consecutive transitions with interval $h = 2$ from both successful and failed trajectories, where the intervals are used to avoid repeated states. Same as next state identification, we also sample steps from steps within $[2, n-2]$ to avoid trivial predictions. For each objective $g$, we annotate pairs where $S^{\text{true}}$ represents a more promising subsequence drawn from a successful trajectory, and $S^{\text{false}}$ represents a less effective subsequence from a failed trajectory. More task details can be found in Appendix A.1.

**Results:** Table 1 shows that all models perform strongly. Claude-3.7-Sonnet achieves the highest accuracy (86.7%), followed by Claude-3.5-Sonnet (85.7%) and Qwen-2.5-VL-72B (83.7%). The consistently high performance across models suggests that LLMs possess reasonable ability to evaluate which transitions are conducive to task progress.

### 3.4 DISCUSSION

Overall, our probing analysis reveals that modern LLMs demonstrate relatively good short-term predictive and local evaluative capabilities: they can reliably identify next states and recognize task-relevant transitions. However, these strengths do not extend to long-horizon planning, where performance deteriorates sharply in aligning its knowledge to specific environments. This suggests that LLMs might inherently lack robust generalization for world modeling across dynamic environments, thus may require external guidance to sustain accurate simulations over extended horizons.

## 4 R-WoM FRAMEWORK

From the probing analysis in Section 3, we identify grounding as a key mechanism for improving the alignment of LLMs to specific environments, which motivates the design of our R-WoM framework.

### 4.1 OVERVIEW

As illustrated in Figure 2, the R-WoM framework employs the retrieval-augmented way to ground world modeling during simulation. Given the task objective and current observation, relevant documentation and tutorials are retrieved and reranked to form the grounding evidence set. This evidence is used to condition the world model during both state transition prediction and reward estimation process. Algorithm 1 summarizes the complete R-WoM pipeline, which iteratively applies this process until task completion or termination.

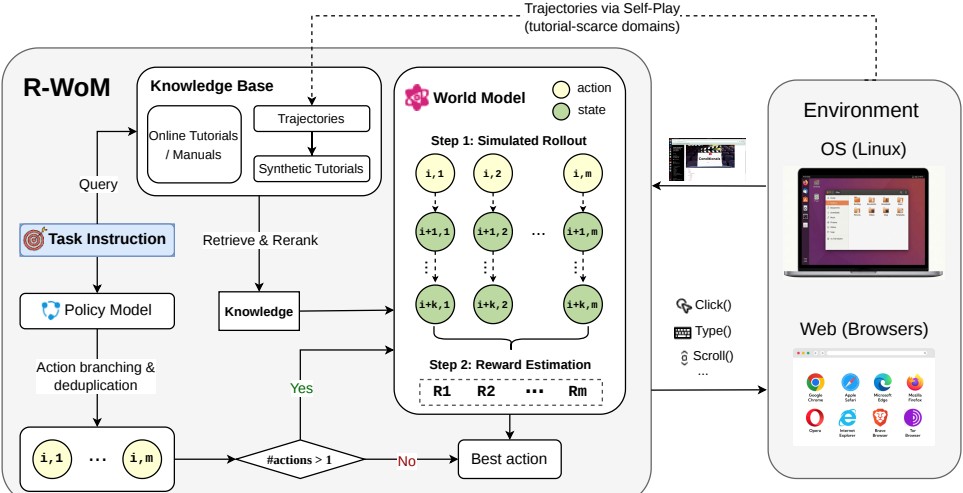

Figure 2: Overview of the R-WoM pipeline. At each time step $i$, the policy model generates $m$ candidate actions. For each candidate, the world model grounded by retrieved tutorials performs $k$-step rollouts to simulate a possible future trajectory. The rewards of rollout trajectories are finally estimated by world models to select the best action.

---

**Algorithm 1** The Pipeline of R-WoM

---

**Require:** Task objective $g$, initial observation $o_1$

1: $\mathcal{E} \leftarrow$ Retrieve and rerank tutorials relevant to the objective $g$
2: $i \leftarrow 1$
3: **while** task not completed **do**
4:      $\mathcal{A}_c \leftarrow \{(t_i^{(1)}, a_i^{(1)}), (t_i^{(2)}, a_i^{(2)}), \ldots, (t_i^{(m)}, a_i^{(m)})\} \sim \pi_p(\cdot|g, o_i)$
5:      **for** each $(t_i^{(j)}, a_i^{(j)}) \in \mathcal{A}_c$ **do**
6:          Generate rollout trajectory $\hat{\tau}_i^{(j)} = \pi_w^{\mathrm{LongCoT}}(o_i, t_i^{(j)}, a_i^{(j)}; \mathcal{E})$
7:      **end for**
         $(t_i^*, a_i^*) = \arg\max_{(t_i^{(j)}, a_i^{(j)}) \in \mathcal{A}_c} \left[ f_w\big(R(\hat{\tau}_i^{(j)}, g, \mathcal{E})\big) \right]$
8:      Execute $a_i^*$, observe $o_{i+1}$
9:      $i \leftarrow i + 1$
10: **end while**

---

### 4.2 Design Details

**RAG design.** We use a reasoning-aware retrieval pipeline to improve the relevance of retrieved tutorials to a given task. The knowledge base contains both online documents and offline trajectories; the latter can be synthesized into experience-style tutorials, which we elaborate in Section 5.5.

Given a task goal $g$, we first encode it into a retrieval query $q = f_{\mathrm{enc}}(g)$. We then retrieve a candidate set $\mathcal{C}_k$ by selecting the top-$k$ chunks under cosine similarity in embedding space. Because embedding similarity alone can miss fine-grained task constraints, we further apply an LLM-based list-wise reranker (implemented with the policy model $p$) to score candidates conditioned on $(q, \mathcal{C}*k)$ and produce a ranked evidence set:

$$\mathcal{E} = f * p^{\mathrm{rank}}(\mathcal{C}_k, q), \tag{9}$$

where $\mathcal{E}$ denotes the final tutorial evidence used by the world model. The world model then conditions on $\mathcal{E}$ to ground future-state imagination and reward estimation.

**R-WoM design.** At step $i$, given tutorial evidence $\mathcal{E}$, we enumerate candidate thought–action pairs $(t_i^{(j)}, a_i^{(j)}) \in \mathcal{A}_c$ and use a world model to simulate their future consequences. For each candidate, we generate a $k$-step imagined trajectory via a LongCoT rollout:

$$\hat{\tau} * i^{(j)} = \pi_w^{\mathrm{LongCoT}}(o_i, t_i^{(j)}, a_i^{(j)}; \mathcal{E}). \tag{10}$$

In contrast to iterative rollout schemes in prior work (Gu et al., 2024; Fang et al., 2025), which require multiple rounds of model calls, we adopt a reasoning-based LongCoT rollout inspired by Deepseek-R1 (Guo et al., 2025). This allows the world model to unfold the entire multi-step imagination trajectory within a single forward reasoning sequence, improving rollout efficiency. To further reduce rollout cost, we adopt an adaptive rollout strategy based on the observation that exhaustive world-model simulation is unnecessary at every step. The design has two components. (1) Adaptive action branching: at step $i$, we prompt the policy to decide how many action candidates to propose, generating $m$ candidates with $1 \leq m \leq n$. The policy expands multiple actions only when it is uncertain about the next move; otherwise it proposes a single high-confidence action. (2) Action deduplication: before launching rollouts, we use the policy itself as a verifier to prune redundant candidates, filtering out actions that are semantically equivalent. More details of the implementations and ablations of the performance-cost tradeoff are in Appendix A.3 and A.4.

Besides, we also observe that absolute sparse rewards that used in prior works (Chae et al., 2024; Gu et al., 2024; Fang et al., 2025) can be insensitive when candidate rollouts are all plausible yet differ in subtle ways. Therefore, inspired by recent progress in relative reward modeling (Liu et al., 2024; Choi et al., 2024; Guo et al., 2025), we apply a list-wise ranking mechanism that compares $\hat{\tau}_i^{(j)}$ and assigns relative preference scores using LongCoT reasoning. Additional ablations on this reward design are provided in Appendix A.6.

$$(t_i^*, a_i^*) \;=\; \arg \max_{(t_i^{(j)}, a_i^{(j)}) \in \mathcal{A}_c} \left[ f_w\big( R(\hat{\tau}_i^{(j)}, g, \mathcal{E}) \big) \right] \tag{11}$$

As is shown in Equation 11, each rollout trajectory is scored relatively in the comparative context of all candidates. In this way, we aim to reduce potential bias from absolute reward signals and stablize the selection of most promising action candidate.

## 5 EXPERIMENT

To evaluate the effectiveness of R-WoM, we propose the following research questions:

- **RQ1**: Does R-WoM improve the performance of computer-use agents compared to established baselines in realistic environments such as browsers and operating systems?
- **RQ2**: How do external tutorials contribute to grounding world models, and to what extent do agents benefit from incorporating this information from tutorials?
- **RQ3**: Can tutorial-grounded world models support longer imagination horizons more effectively than ungrounded counterparts over multi-step rollouts?
- **RQ4**: Whether R-WoM can be extended to scenarios where existing tutorials are scarce?

### 5.1 SETUP

We evaluate R-WoM against three baselines:

- **Vanilla**: The vanilla approach is adapted from the official implementations: the screenshot-only version for OSWorld provided by GTA-1 (Yang et al., 2025), and the screenshot+accessibilty tree version for WebArena provided by WMA (Chae et al., 2024). This approach relies solely on the task objective, current observation and prior interaction history.
- **RAG**: A retrieval-augmented generation pipeline that retrieves relevant documentation and augments the LLM before action prediction, which is built upon the vanilla approach.
- **WebDreamer**: A world model method proposed by Gu et al. (2024) where it adopts an iterative way of generating rollouts through the communication between policy model and world model.

We conduct experiments on two challenging online environments: **WebArena** (Zhou et al., 2023), which spans web-based tasks across domains (e.g., e-commerce, social forums); and **OSWorld** (Xie et al., 2024), which covers tasks in various desktop applications (e.g., chrome, gimp, libreoffice). Specifically, we sample a subset from these two benchmarks (87 from 361 for OSWorld and 113 from 301 unique templates for Webarena) for our experiments where tutorials are available for

Table 2: End-to-end performance on OSWorld and WebArena across three runs. Best in **bold**; second-best underlined. ↑ denotes this relative improvement over the second-best baseline

| Model | Method | OSWorld (Xie et al., 2024) | WebArena (Zhou et al., 2023) |
|---|---|---|---|
| Qwen-2.5-VL-72B | Vanilla | $26.36 \pm 2.32$ | $21.84 \pm 0.42$ |
| | RAG | $\underline{30.84 \pm 1.07}$ | $22.42 \pm 0.42$ |
| | WebDreamer | $28.37 \pm 2.01$ | $\underline{24.50 \pm 0.84}$ |
| | **R-WoM** | $\mathbf{37.48 \pm 2.29} \uparrow_{21.5\%}$ | $\mathbf{28.49 \pm 0.43} \uparrow_{16.3\%}$ |
| Claude-3.5-Sonnet | Vanilla | $22.43 \pm 2.25$ | $27.74 \pm 0.43$ |
| | RAG | $22.19 \pm 0.92$ | $\underline{30.70 \pm 0.41}$ |
| | WebDreamer | $\underline{23.48 \pm 2.14}$ | $29.82 \pm 0.41$ |
| | **R-WoM** | $\mathbf{26.01 \pm 0.44} \uparrow_{10.8\%}$ | $\mathbf{33.15 \pm 0.01} \uparrow_{8.0\%}$ |
| Claude-3.7-Sonnet | Vanilla | $28.47 \pm 2.27$ | $28.92 \pm 0.41$ |
| | RAG | $27.76 \pm 0.75$ | $\underline{32.75 \pm 0.72}$ |
| | WebDreamer | $\underline{31.24 \pm 2.88}$ | $31.86 \pm 0.01$ |
| | **R-WoM** | $\mathbf{38.54 \pm 1.92} \uparrow_{23.4\%}$ | $\mathbf{34.58 \pm 1.10} \uparrow_{5.6\%}$ |

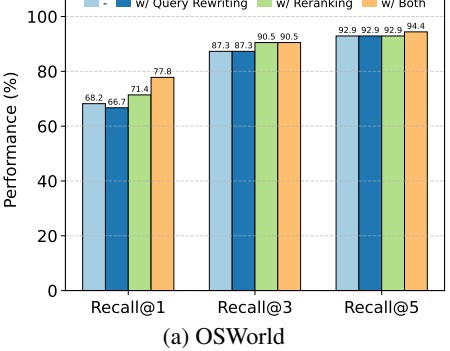

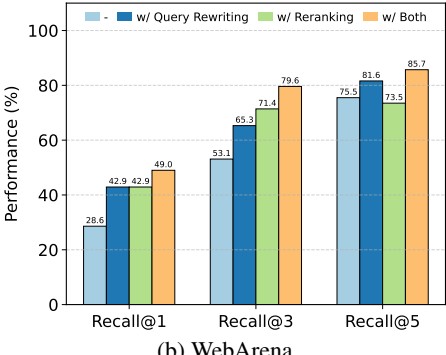

(a) OSWorld (b) WebArena

Figure 3: Retrieval performance under different retrieving strategies.

retrieval purpose and we collect tutorials from both online websites. Besides, we also extend R-WoM to tutorial-scarce scenarios by synthesizing tutorials from self-played trajectories (report in Section 5.5). The details of the subsets and tutorial collection can be found in Appendix A.2. If not specifically mentioned, our evaluations are conducted on the following models: **Qwen-2.5-VL-72B (Instruct version)** (Bai et al., 2025), **Claude-3.5-Sonnet**, and **Claude-3.7-Sonnet**, serving as both the policy and world model. For methods requiring retrieval, we build the RAG pipeline with Langchain[4], FAISS (Douze et al., 2024) as the vector store, and Qwen-3-Embedding-8B (Zhang et al., 2025b) as the embedding model. More implementation details can be found in Appendix A.3.

## 5.2 RQ1: END-TO-END PERFORMANCE

Table 2 reports the overall end-to-end performance. It shows that R-WoM consistently outperforms all alternatives, with improvements of +21.5% on OSWorld and +16.3% on WebArena for Qwen-2.5, +10.8% and +8.0% for Claude-3.5, and +23.4% and +5.6% for Claude-3.7 over the strongest non-R-WoM baselines. These results reveal that the improvements remain stable across different backbones, highlighting that R-WoM provides more consistent benefits compared with retrieval alone or ungrounded world modeling. More detailed results of breakdown in domains and failure mode analysis can be found in Appendix A.5.

## 5.3 RQ2: THE ROLE OF TUTORIALS IN GROUNDING WORLD MODELS

In this section, we evaluate how grounding quality, ranging from no retrieval, to the retrieval R-WoM by default use, then to oracle-level retrieval of tutorials translates into end-to-end task success. To have a better view of the retrieval quality used in R-WoM, we first evaluate the performance of the retrieval component in R-WoM under different configurations. The results in Figure 3 shows that retrieval recall improves most when query rewriting and reranking are used together, showing that

---
[4] https://github.com/langchain-ai/langchain

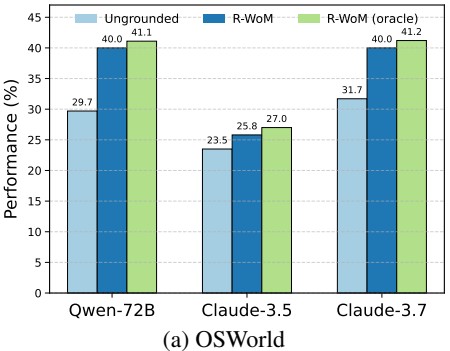 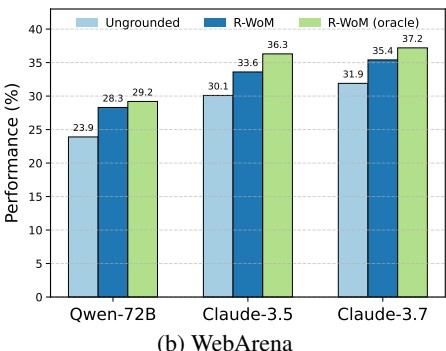

(a) OSWorld         (b) WebArena

Figure 4: Performance under different grounding settings, where we compare ungrounded world model: WebDreamer, world model grounded with retrieved tutorials: R-WoM, and world model grounded with oracle tutorials: R-WoM (oracle).

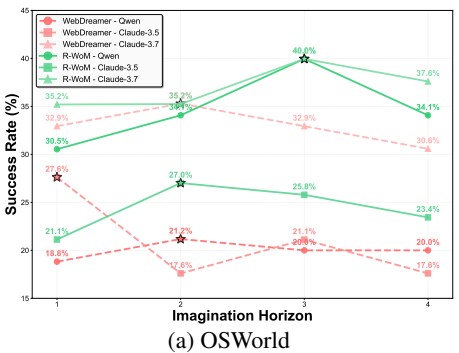 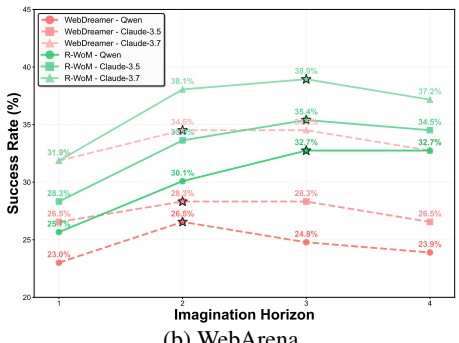

(a) OSWorld         (b) WebArena

Figure 5: Success rates (%) across imagination horizons on OSWorld (a) and WebArena (b). R-WoM (green, solid) consistently outperforms WebDreamer (red, dashed) and reaches its peak at larger imagination horizon (at horizon around 3).

these techniques are complementary. Query rewriting shows benefits when task phrasing is vague (e.g., Fork ChatGPT). In contrast, reranking offers benefits across both benchmarks by filtering out semantically irrelevant candidates. Overall, the results show that the retrieval can reach over 85% and 90% recall@5, respectively. Examples of the retrieved content can be seen in Appendix A.5. To further analyze how retrieval fidelity impacts the model's reasoning quality, we compare three grounding settings: (i) no grounding (i.e., WebDreamer), (ii) grounding with R-WoM using retrieved tutorials, and (iii) grounding with R-WoM using tutorials under oracle retrieval (human annotated). The oracle tutorial setting is conducted by manually annotating one most relevant tutorial from the knowledge base to each specific task, similar as what is employed in Section A.1. As shown in Figure 4, performance improves monotonically with the grounding quality, from no external knowledge to retrieved tutorials, and finally to oracle retrieved tutorials. This trend indicates that the accuracy of procedural knowledge effectively contributes to long-horizon simulation.

### 5.4 RQ3: ABLATION STUDIES OF IMAGINATION HORIZON

To study the effect of imagination horizon on end-to-end performance, we vary the horizon from 1 to 4 for both ungrounded (WebDreamer) and grounded (R-WoM) world models. Figure 5 shows that, WebDreamer, the world model without grounding during rollouts, shows initial gains but quickly plateaus and even declines beyond 2 steps, reflecting its susceptibility to compounding prediction errors. In contrast, R-WoM maintains consistently higher success across horizons on both OSWorld and WebArena, with improvements lasting up to horizon three before decreasing. These results suggest that tutorial-guided grounding helps stabilize rollouts over longer horizon simulations.

### 5.5 RQ4: EXTENSION TO SCENARIOS WHERE EXISTING TUTORIALS ARE SCARCE.

To further investigate how R-WoM performs in scenarios where it is hard to find existing tutorials, we extend R-WoM to these scenarios by synthesizing tutorials directly from self-played trajectories,

Table 3: Performance comparison on OSWorld tasks under tutorial-scarce settings.

| Model | Claude-3.7-Sonnet | Claude-4-Sonnet | Claude-4.5-Sonnet |
|---|---|---|---|
| Vanilla | 32.25% | 35.82% | 45.83% |
| RAG | 33.36% | 36.93% | 46.11% |
| WebDreamer | 30.86% | 34.43% | 46.35% |
| **R-WoM** | **35.71%** | **39.28%** | **49.29%** |

inspired by recent works leveraging procedural memory from self-play (Wang et al., 2024b; 2025c). Specifically, we utilize 2k open-sourced trajectories released in AgentNet (Wang et al., 2025b) and synthesized approximately 1.3k synthesized tutorials that could be useful to the tasks of OSWorld. The details of our synthesis pipeline can be found in Appendix A.2. It is important to mention that these trajectories do not have task overlap with our test tasks. Then these synthesized tutorials serve as general operation guidelines for tasks lacking online references during our evaluation on three Claude-Sonnet models (3.7, 4 and 4.5). As shown in Table 3, R-WoM has consistent improvement over other baselines across these three models. This demonstrates that R-WoM can adapt to tutorial-scarce tasks by grounding the world model using synthesized tutorials derived from self-play.

## 6 RELATED WORKS

### 6.1 COMPUTER-USE AGENT

One line of works focuses on exploring how to improve agent's understanding of computer-use actions, such as building end-to-end agent frameworks (Agashe et al., 2024; 2025; Song et al., 2025; Mei et al., 2025), and training native agent models (Qin et al., 2025; Wang et al., 2025a; Lai et al., 2025) or specific action grounding models (Wu et al., 2024; Xie et al., 2025; Yang et al., 2025). Another line of works explores treating LLMs as world models to simulate the computer-use environments. WebDreamer (Gu et al., 2024) pioneers this direction by using LLMs to simulate the outcome of candidate actions, and evaluate these imagined states with discrete reward given by LLM judge (Gu et al., 2024). Subsequent works such as WMA (Chae et al., 2024) adapt this idea to improve planning by abstracting state transitions into natural language summaries. WKM (Qiao et al., 2024) and WebEvolver (Fang et al., 2025) develop co-evolving world models and policies to progressively refine both simulation and planning, moving beyond one-horizon imagination.

### 6.2 TUTORIAL-USE

Parallel developments leverage tutorials or indirect knowledge to train digital agents. Synatra (Ou et al., 2024) converts human-oriented tutorials into 100k synthetic demonstrations to fine-tune a 7B CodeLLaMA model. Other frameworks generate trajectories guided by tutorial completion or replay (e.g., AgentTrek (Xu et al., 2024), TongUI (Zhang et al., 2025a)) to teach GUI navigation and tool use from multimodal resources. Learn-by-interact (Su et al., 2025) synthesizes trajectories by leveraging tutorials and interaction with the environments. These approaches focus on offline trajectory generation by referring to tutorials while our approach focuses on tutorial-guided grounding of LLMs as world models at inference time.

## 7 CONCLUSION AND FUTURE WORK

We presented a systematic study of LLM-based world models for computer-use tasks, revealing that while they can model state transitions and recognize task-relevant progress, they fail to reliably adapt to unfamiliar environments in long-term planning without grounding. To address this, we proposed the Retrieval-augmented World Model (R-WoM), which incorporates environment-specific tutorial knowledge during the imagination rollouts and reward prediction procedures to reduce hallucinations and stale knowledge. Evaluations on WebArena and OSWorld show that R-WoM consistently outperforms competitive baselines, demonstrating the efficacy of retrieval-augmented grounding for LLM agents in dynamic browser-use and computer-use scenarios.

## 8 ETHICS AND REPRODUCIBILITY STATEMENT.

**Ethics.** Our work uses only publicly available benchmarks (OSWorld and WebArena). While retrieval-augmented methods may inherit biases from external sources, our study remains confined to controlled environments.

**Reproducibility.** Our work is reproducible. We provide the algorithm process of our method, Retrieval-augmented World Model (R-WoM), in Algorithm 1. The experimental setup, are described in Section 5.1 and the implementation details are provided in Appendix A.3.

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

# A APPENDIX

**Roadmap**: Section A.1 introduces the design details of our probing task. Section A.2 introduces the tutorial collection, annotation and retrieval approach for our experiments. Section A.3 presents the implementation details of R-WoM, including action space definition and prompt design. Section A.4 presents our discussion of cost-performance tradeoff and our further cost optimization of R-WoM. Section A.5 shows our deep analysis of the failure cases of R-WoM. Section A.6 shows more results of our ablation studies.

## A.1 DETAILS OF PROBING TASK

---

**PROMPT FOR NEXT STATE IDENTIFICATION**

Given the previous state of the web page: {previous_state} and the current action: {current_action}, please reason about the next state. The next state can be one of the following: {state_a}, {state_b}. Please reason about the next state and return the rationale and the choice. The choice should be one of the following: A, B. Output the choice in the following JSON format:

```
{
    "rationale": "...",
    "choice": "..."
}
```

---

**Task 1: Next-state identification.** To assess whether the world model can predict the immediate outcome of an action given the current state, the model is asked to discriminate between the true next observation and a lexically similar distractor, as illustrated in Figure 6. In this way, we aim to probe LLM's sensitivity to environment changes. We construct 100 samples drawn from trajectories in WebArena for this task.

**Task 2: Full-procedure planning alignment.** Moving beyond identifying next state, we would like to probe whether LLM can reason about longer steps of future states. As shown in A.1, given a task objective, the model is asked to generate a multi-step plan, which is then validated against tutorials describing environment dynamics. The evaluation measures whether the model's procedure aligns with realistic element locations, operation sequences, and interaction methods. To assess this capability, we construct 40 samples from trajectories in both OSWorld and WebArena.

---

**PROMPT FOR FULL-PROCEDURE PLANNING ALIGNMENT**

You are a grounding validation assistant that verifies whether tutorial-referenced operations in a plan are accurately grounded in the provided documentation.

**Evaluation criteria**
- Element Text Accuracy: Exact text matches between plan and tutorial for referenced elements.
- Location Consistency: Location indicators (position, context) align with tutorial descriptions.
- Operation Sequence: Prerequisites and dependencies match tutorial methodology.
- Interaction Method: Specified actions (click, input, select) align with tutorial instructions.
- Attribute Precision: Element types, properties, and characteristics match tutorial specifications.

**Evaluation principle**
- Accept: Plan steps that extend beyond tutorial scope (additional operations are allowed).
- Reject: Any tutorial-referenced operation with misaligned text, location, or method.

**Output Format**
Output your response in the following JSON format:

```
{
    "rationale": "Your rationale of your evaluation",
    "answer": "yes/no"
}
```

---

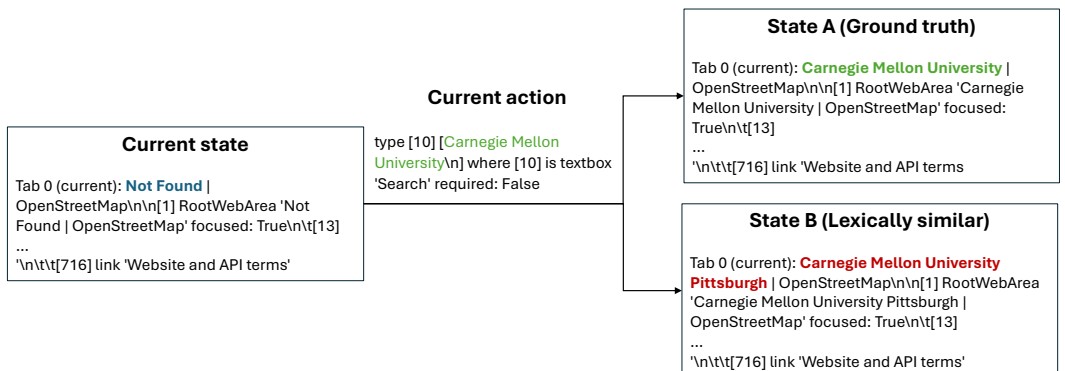

Figure 6: Illustration of the **next-state identification** probing task. Given a current state and an action, the model must choose between two candidate next states: (A) the ground-truth state, and (B) a lexically similar distractor. This task evaluates whether the world model can correctly predict the true next observation rather than being misled by textual similarity.

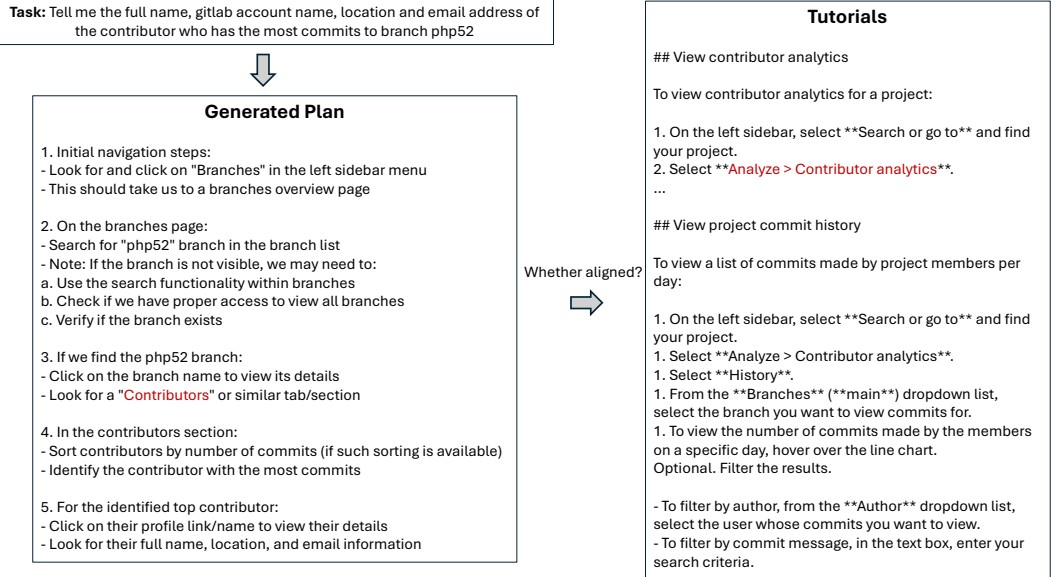

Figure 7: Illustration of the **full-procedure planning alignment** probing task. Given a task objective (top), the model generates a multi-step plan (left), which is then compared against environment-specific tutorials (right). The evaluation checks whether the generated procedure aligns with the tutorials in terms of navigation logic, element selection, and operation feasibility. This task assesses the world model's ability to sustain long-horizon procedural reasoning in realistic environments.

---

**PROMPT FOR MILESTONE TRANSITION RECOGNITION**

You are evaluating web automation trajectories to identify which one is more likely to succeed in completing the given task.

The following two trajectories show segments from different agent attempts at the same task. Both agents were following the same initial steps, but diverged when they chose different actions at a critical decision point. Your task is to determine which trajectory segment demonstrates better progress toward completing the task objective. You need to output in the following JSON format as:

```
{
    "answer": "A/B",
    "rationale": "xxx"
}
```

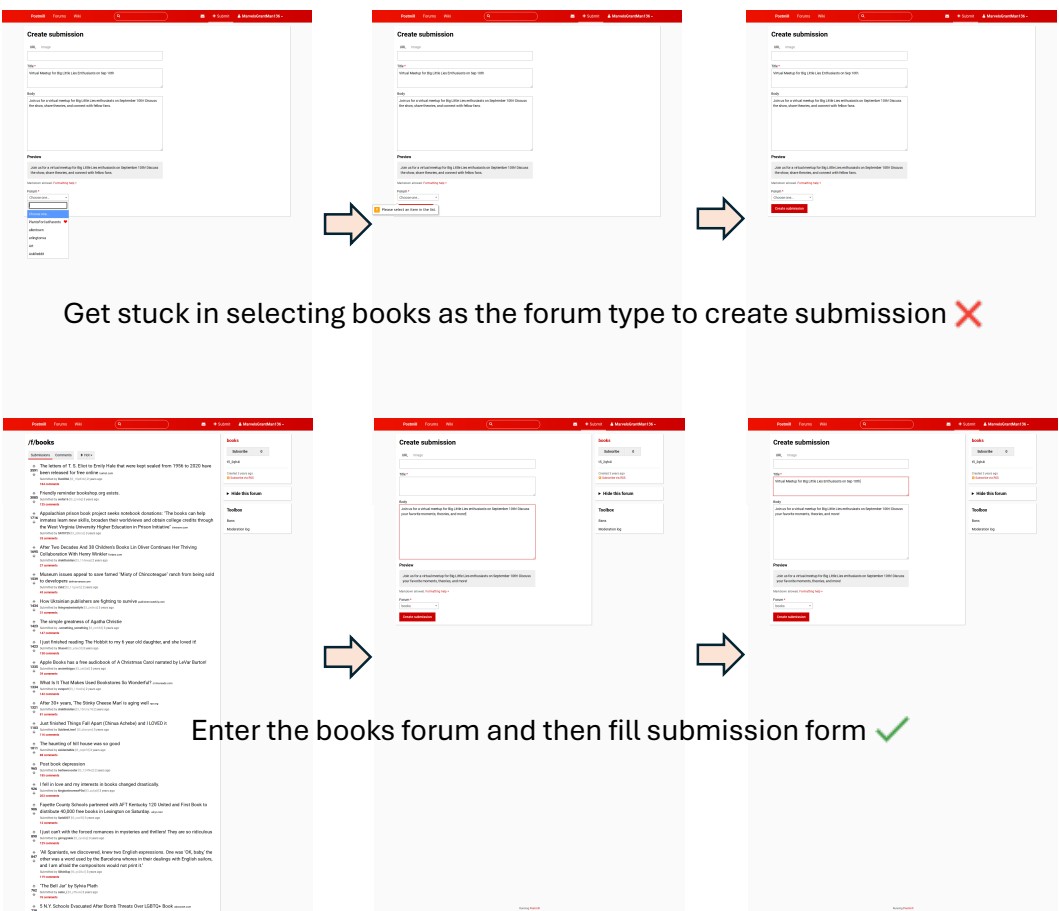

Figure 8: Illustration of the **milestone transition recognition** probing task. Given a sequence of transitions, the model must identify whether they reflect meaningful progress toward the goal. In this example, the top path shows an unproductive transition where the agent gets stuck trying to directly select "books" as a forum type, failing to proceed. The bottom path shows a more promising milestone transition: the agent first enters the books forum and then successfully fills out the submission form. The task evaluates whether the world model can distinguish between effective and ineffective procedural progress.

**Task 3: Milestone transition recognition.** To probe reward estimation capability of LLMs, we design this task to assess whether LLMs have the capability to capture meaning state transitions. As shown in Figure 8, the LLM is presented with pairs of trajectory segments that diverge at a decision point, one representing a promising milestone transition and the other an unproductive path. The LLM needs to identify which trajectory is more conducive to task success. This setting is evaluated on 98 samples drawn from both successful and failed trajectories in WebArena.

## A.2 TUTORIAL PROCESSING

Our framework relies on tutorials as external grounding for browser- and computer-use tasks. To construct a comprehensive knowledge base, we gather tutorials from both general-purpose and environment-specific resources. For cross-domain instructional guidance, we include WikiHow, which provides structured, step-by-step content spanning a broad range of tasks. For environment-specific domains, we incorporate official documentation from the corresponding software or websites. The complete list of tutorial sources is as follows:

- WikiHow: `https://www.wikihow.com/Main-Page`
- Google Chrome Help: `https://support.google.com/chrome`

- GIMP 3.0 User Manual: `https://docs.gimp.org/3.0/en/`
- Visual Studio Code Documentation: `https://code.visualstudio.com/docs`
- Ubuntu Help: `https://help.ubuntu.com/22.04/ubuntu-help/`
- Mozilla Thunderbird Support: `https://support.mozilla.org/en-US/products/thunderbird/learn-basics-get-started`
- VLC Media Player User Guide: `https://docs.videolan.me/vlc-user/desktop/3.0/en/`
- LibreOffice Help: `https://help.libreoffice.org/latest/en-US/`
- GitLab Documentation: `https://docs.gitlab.com/`
- Adobe Commerce Admin User Guides: `https://experienceleague.adobe.com/en/docs/commerce-admin/user-guides/home`

From these sources, we construct a knowledge base of over 30k chunked tutorial documents that collectively support tasks across diverse software and website environments. Since our framework requires tutorial availability to provide concrete grounding, we sample task subsets from OSWorld and WebArena that can be partially mapped to tutorial examples. Specifically, in OSWorld, 85 tasks have clear and verifiable tutorial references from official online documents covering domains such as Chrome, GIMP, VSCode, VLC, Thunderbird, while 276 tasks lack such references. For WebArena, based on its 301 unique task templates, 113 have tutorial coverage covering CMS and GitLab domains and 188 do not. This split is determined using a consistent criterion: whether a task's goal can be matched to explicit operation instructions from official online tutorials or offline documentation for the target software or website. We annotate one document chunk for each task from human's perspective as the oracle tutorial for each task.

**Synthesis of tutorials.** In domains where it is hard to find online/offline tutorial references, we extend R-WoM by adopting a tutorial synthesis approach. Inspired by Wang et al. (2025c), we employ a two-stage synthesize-then-consolidate pipeline to generate tutorials from self-played trajectories released by AgentNet (Wang et al., 2025b). Specifically, we first generate skill-level tutorials from each trajectory and then we group synthesized tutorials by domain and vector similarity to further conduct a consolidation including deduplication and merging. The prompts we use are as below.

---

**PROMPT FOR SYNTHESIZING TUTORIALS FROM TRAJECTORIES**

You are a helpful assistant that synthesizes skill-level tutorials from a trajectory of observations and actions in a computer-use environment.

**Checklist**

- Make the tutorial more general and deanonymize the task.
  - For example, a "readme.txt" file should be replaced with a ".txt format file".
- Make the operations more general and avoid specific values unless the specific values are system settings or preferences.
  - For example, instead of "Fill row 7 by summarizing the values range from row 1 to row 6," use a generalized Excel-style operation such as "Fill a target cell range by applying a summary formula (e.g., =AVERAGE([MASK]:[MASK])) to another cell range."
- Pay attention to potential blockers in the trajectory.
  - For example, if the trajectory is stuck in a loop, you should mention it in the tutorial to help avoid this.
- When generating solutions, carefully observe the initial state to avoid unnecessary operations.
  - If the task is to search information or in a specific website (e.g., www.amazon.com), assume that Chrome is already open and the user is already on the website.
  - If the task is to edit a specific file (e.g., readme.txt), assume that the file is already open and already contains existing content.

**Action space:**

- left_click, right_click, middle_click, double_click, left_click_drag, type, scroll, key

**Output format**

---

- Organize and abstract the operations as skills. Try to extract all the related skills from the given trajectory.
- Each skill must contain 3–6 concrete actions that are defined in the action space, meaning the skill should not be too simple or too complex.
- Avoid unnecessary actions such as closing a window or verifying command execution success.

An example of a skill:

```
<skill>Open the settings page in Chrome browser
<prerequisites>
- The Chrome browser is open and the homepage of amazon.com is
    loaded.
</prerequisites>
<actions>
- left_click on the three dots icon in the top right corner of the
    browser.
- left_click the settings option to open the settings page.
</actions>
</skill>
```

---

### PROMPT FOR CONSOLIDATING TUTORIALS

You are a helpful assistant that merges similar skills from the given documents.
**Action space:**

- left_click, right_click, middle_click, double_click, left_click_drag, type, scroll, key

**Output format**

- If multiple skills represent similar operations, merge them into a single skill to avoid duplication.
- If multiple skills achieve the same goal through different methods or paths, combine them into one skill and describe the alternative approaches within it.
- After merging, make sure each skill still maintains the original format with `<prerequisites>` and `<actions>` tags.
- Carefully verify that no duplicate skills remain and that all unique skills are retained.

Output all qualified skills in the following format:

```
<skill>skill 1</skill>
<skill>skill 2</skill>
...
<skill>skill N</skill>
```

---

To retrieve useful tutorials at inference time, we adopt a reasoning-based retrieval strategy. This involves query rewriting to anonymize and generalize task queries, followed by LLM-based reranking to reduce false negatives that may arise when relying solely on cosine similarity. The detailed prompts used for query rewriting and reranking are provided below.

---

### PROMPT FOR QUERY REWRITING

You are an AI assistant that rewrite original query into comprehensive, searchable queries that are easier to retrieve answers from documents. You must follow these rules:

- Organize the original query to be well-structured and clear with details: Try to make the query detailed and clear. For example, instead of a title like "Fork ChatGPT", a good rewritten query would be, "How could I fork the ChatGPT repository in the gitlab?"
- Generalize Personal Details: Replace all specific, personal information (like user names, file names, file location) with general descriptions (like "a user", "a xxx format file", "at desktop").

Table 4: Action space for WebArena and OSWorld.

| Environment | Action | Definition |
|---|---|---|
| **WebArena** | `click` | Clicks a webpage element identified by its id. |
| | `type` | Types text into a webpage element; may submit if appropriate. |
| | `hover` | Moves the cursor over a webpage element. |
| | `press` | Presses a key or key combination. |
| | `scroll` | Scrolls the page up or down. |
| | `new_tab` | Opens a new browser tab. |
| | `tab_focus` | Focuses a specific browser tab. |
| | `close_tab` | Closes the active browser tab. |
| | `goto` | Navigates the current tab to a URL. |
| | `go_back` | Navigates to the previous page. |
| | `go_forward` | Navigates to the next page. |
| | `stop` | Terminates the task and returns an answer (use `N/A` if unknown). |
| **OSWorld** | `left_click` | Left-clicks a described UI element in the desktop environment. |
| | `right_click` | Right-clicks a described UI element in the desktop environment. |
| | `middle_click` | Middle-clicks a described UI element in the desktop environment. |
| | `double_click` | Double-clicks a described UI element in the desktop environment. |
| | `triple_click` | Triple-clicks a described UI element in the desktop environment. |
| | `left_click_and_drag` | Drags from one described UI location to another. |
| | `key` | Presses a list of keys. |
| | `scroll` | Scrolls within a described element. |
| | `hcroll` | Scrolls within a described element horizontally. |
| | `type` | Types text into a described element. |
| | `wait` | Pauses execution for a short duration. |
| | `done` | Ends the task successfully and returns the final answer if any. |
| | `fail` | Ends the task with failure and stop. |

---

**PROMPT FOR RERANKING**

Your task is to re-rank a list of documents based on their relevance to a given task. Carefully analyze the task and each numbered document. Your goal is to identify which documents are helpful for completing the task and order them accordingly.
Your output must be a single JSON object with one key: "reranked_indexes". The value for this key must be a list of the original document indexes, sorted from most relevant to least relevant.
**Example format:**

```
{
    "reranked_indexes": [0, 2, 1]
}
```

---

## A.3 IMPLEMENTATION DETAILS OF R-WOM

To enable automation in browser and computer-use environments, we adopt the official action space definitions provided by WebArena[5] and OSWorld[6], as summarized in Table 4. In practice, we find that direct action coordinate mapping in OSWorld poses challenges for models such as the Qwen series and Claude-3.5-Sonnet. To address this and enable the policy model to generate more effective actions during world model rollouts, we employ GTA-1-7B (Yang et al., 2025) as an auxiliary action grounding model to assist in action generation when evaluating on OSWorld. For retrieval-related approach (i.e., RAG and R-WoM), we use top-5 retrieved document chunks by default to put them into the LLM's context.

**PROMPT FOR GENERATING ACTION CANDIDATES (ADAPTIVE BRANCHING)**

You are a reasoner that analyzes the current state, previous actions, and task progress to determine the next required action.

---

[5]`https://github.com/web-arena-x/webarena`
[6]`https://github.com/xlang-ai/OSWorld`

**Available actions**
# Action space definition
**Rules for success**

- When pressing keys, ensure held/pressed keys are within {`KEYBOARD_KEYS`}.

- Output a single action at each step; do not bundle multiple intents into one step.

- Only issue actions that are valid for the current observation (e.g., do not type into buttons or click static text).

- Strictly avoid repeating the same action if the interface state is unchanged.

**Response JSON schema**

```
{
  "observation": "Description of current state and any changes
      observed",
  "action_candidates": [
    {
      "thought_and_action": "Why this action is appropriate given
          the observation",
      "action_code": {
        "action_type": "action_type",
        "parameters": {
          "param1": "value1",
          "param2": "value2"
        }
      }
    }
  ]
}
```

**Output requirements**

- Observation: First describe the current environment state changes:
    - Wrap the content in `<observation> ... </observation>`.
    - Focus on what has changed from the previous state; keep it concise.
    - Do *not* describe intended actions in this section.
- Action candidates: Then propose **1 to {ACTION_N}** candidate next actions:
    - Propose **1** candidate when progress is on track and the next step is unambiguous.
    - Propose **2 to {ACTION_N}** candidates *only* when the agent is stuck or previous actions failed. Candidates must be diverse (e.g., different interaction types or different target elements), not minor variants of the same attempt.
    - Wrap the full list in `<action_candidates> ... </action_candidates>`.
    - Order candidates by confidence (most confident first).
    - Each candidate must include:
        * *thought*: wrap in `<thought> ... </thought>`, describing the rationale/intent.
        * *action*: wrap in `<action> ... </action>`, containing **exactly one line** of Python code.
    - The action code must:
        * Use the `aci` class.
        * Be enclosed in ```` ```python ```` fences.
        * Contain exactly one executable action line (e.g., `aci.left_click([100, 100])`).

## PROMPT FOR DEDUPLICATION

You are a helpful assistant that assesses whether a list of proposed action candidates contains meaningful diversity. Your goal is to determine whether invoking a **World Model (WM)** to simulate and rerank candidates is necessary.

**When WM is NOT needed (output `NO`).**

- **Minor parameter variance:** candidates execute the same action on the same logical target, differing only in non-semantic details (e.g., small coordinate shifts or pixel-level variations).

- **Trivial reordering:** candidates are permutations of independent steps in the same workflow (e.g., filling `Name` then `Email` vs. `Email` then `Name`).

**When WM is needed (output `YES`).**

- **Semantic diversity:** candidates propose substantively different strategies for the current sub-goal (e.g., using a search bar vs. navigating via category links).

- **Distinct targets:** candidates interact with different UI elements that lead to different execution paths or outcomes.

**Output format**

```
<judgment>YES or NO</judgment>
<rationale>Brief rationale for the judgment.</rationale>
```

---

### PROMPT FOR RETRIEVAL-AUGMENTED FUTURE STATE ROLLOUTS

You are a world-model assistant with extensive knowledge of desktop and web UIs. Given the task objective, and a candidate action and the observation before the action candidate, you must "simulate the future" and describe the plausible future states.

**Available actions**
# Action space definition
**Tutorial usage guideline**

- Use tutorials to identify efficient workflow patterns that should be predicted as likely outcomes.

- Provide a reference to the tutorial if the current situation matches the standard operations in the tutorials. If the current situation does not align with tutorials, rely on internal world knowledge instead.

**Environment awareness checklist**

- Visible UI elements: text, icons, menus, modals, tooltips

- Element states: enabled/disabled, focused/hovered, loading progress

- Hidden or off-screen affordances revealed by scrolling or clicking

- Cursor position, caret position, selection highlights

- Global context: file system changes, network requests, OS dialogs

**Output Format**
Produce an ordered chain from **STATE 0** (current) up to **STATE n** ($1 \leq n \leq \{k\}$); you may stop early if no further prediction is useful.

---

### PROMPT FOR RETRIEVAL-AUGMENTED REWARD ESTIMATION

You are an agent that evaluates actions by considering the observation before the action candidates and the potential outcomes of these actions.
**Tutorial Grounding Guidance**
Priorize action sequences that follow the standard operations in the tutorials and have captured the milestones and conditions to make more meaningful progress to achieve the task objective.
**Output Format**
Output your response in the following JSON format:

```
{
  "ranking": [x, x, x] # "indexes of the action candidates, most
     promising first",
  "thought": "your rationale for the ranking result"
}
```

---

## A.4 COST-PERFORMANCE TRADEOFF

To better understand the cost of R-WoM lies in retrieval or generation, we analyze the computational overhead and report the average retrieval time per task sample (including query rewriting and reranking) and the average total end-to-end execution time per task on the OSWorld benchmark in Table 5. The results show that retrieval contributes less than 2% of the total latency, confirming

Table 5: Average retrieval and overall execution time per task sample on the OSWorld benchmark.

| Model | Avg retrieval time per task sample | Avg total execution time per task sample |
|---|---|---|
| Qwen2.5-VL-72B-Instruct | 5.1s | 989s |
| Claude-3.5-Sonnet | 5.3s | 518s |
| Claude-3.7-Sonnet | 4.7s | 410s |

Table 6: How R-WoM (Adaptive) performs compared to other methods on OSWorld.

| Model | Claude-3.7-Sonnet | Claude-4-Sonnet | Claude-4.5-Sonnet |
|---|---|---|---|
| Vanilla | 29.40% | 48.24% | 59.12% |
| RAG | 27.76% | 50.82% | 60.43% |
| WebDreamer | 31.24% | 49.65% | 62.09% |
| R-WoM | **39.13%** | **56.73%** | **67.84%** |
| R-WoM (Adaptive) | 37.80% | 55.18% | 66.37% |

Table 7: Efficiency analysis of adaptive mechanisms in R-WoM.

| Model | Adaptive Action Branching | Action Deduplication | WM Trigger Reduction |
|---|---|---|---|
| Claude-3.7-Sonnet | 30.8% | 31.5% | 80.5% |
| Claude-4-Sonnet | 29.7% | 34.2% | 82.2% |
| Claude-4.5-Sonnet | 28.4% | 38.1% | 84.4% |

Table 8: Token usage comparison, where 3.7, 4 and 4.5 all represent Claude-Sonnet models.

| Model | 3.7 Input | 3.7 Output | 4 Input | 4 Output | 4.5 Input | 4.5 Output |
|---|---|---|---|---|---|---|
| Vanilla | 32.45M | 0.62M | 29.91M | 0.37M | 29.35M | 0.36M |
| RAG | 34.39M | 0.66M | 31.69M | 0.38M | 31.10M | 0.38M |
| WebDreamer | 226.37M | 4.35M | 208.60M | 2.56M | 204.69M | 2.51M |
| R-WoM | 76.19M | 1.47M | 70.27M | 0.86M | 68.95M | 0.85M |
| R-WoM (Adaptive) | 35.39M | 0.85M | 32.60M | 0.50M | 31.94M | 0.49M |

that the main computational bottleneck comes from the multi-step generation process rather than retrieval itself.

To analyze the effectiveness of cost optimization of R-WoM, we conduct ablations by removing the adaptive action branching and action deduplication on OSWorld with Claude-3.7-Sonnet, Claude-4-Sonnet, and Claude-4.5-Sonnet. On the performance side, Table 6 shows that R-WoM retains most of the performance advantages of the R-WoM with complete world model triggering while significantly reducing redundant computation. For example, with Claude-4.5-Sonnet, R-WoM reaches 66.37%, which is a 12% relative gain over Vanilla and a 10% gain over RAG, while remaining close to the full R-WoM result of 67.84%. We further measure how much the optimization reduces unnecessary branching and world-model triggers.

On the cost side, Table 7 shows that the adaptive version of R-WoM reduces multi-action generation to about 30% of the original, removes 30-40% of duplicated triggers, and lowers world-model activation to around 15-20% of the initial rate. As shown in Table 8, compared to full R-WoM, the adaptive variant reduces token usage by more than 50% in most settings. Its total token cost is close to RAG, especially for Claude-4.5-Sonnet, where the difference is within 5–10%. These results show that adaptive R-WoM maintains most of the performance improvements of R-WoM while substantially reducing computational cost, achieving a more efficient balance between reasoning strength and inference overhead.

Table 9: Examples of successful and failed retrieval cases.

| Case Type | User Query | Retrieved Tutorial |
|---|---|---|
| Successful | "Computer, can you turn the webpage I'm looking at into a PDF file and put it on my main screen?" | "Print from Chrome — On your computer, open Chrome. Open the page, image, or file you want to print. Click `File` → `Print`, or use `Ctrl+P` / `Cmd+P`. Select 'Save as PDF' and click Print..." |
| Failure | "Please calculate the ages of the employees according to their birthday." | "When entering a time, separate time elements with colons. To change the date or time format in Calc, open Format Cells and select Date/Time..." |

Table 10: Failure statistics of WoM w/o world model (WebDreamer) and w/ world model (R-WoM) across OSWorld and WebArena.

| Benchmark | Model | Retrieval Failures | Task Success Despite Retrieval Failure | Task Failure (WebDreamer Failed as well) |
|---|---|---|---|---|
| OSWorld | Qwen2.5-72B | 5 | 3 | 2 |
| | Claude-3.5-Sonnet | 4 | 3 | 1 |
| | Claude-3.7-Sonnet | 4 | 3 | 1 |
| WebArena | Qwen2.5-VL-72B | 17 | 11 | 6 |
| | Claude-3.5-Sonnet | 17 | 12 | 5 |
| | Claude-3.7-Sonnet | 16 | 11 | 5 |

## A.5 FAILURE MODE ANALYSIS.

To diagnose the failure cases of R-WoM, we conduct a qualitative and quantitative analysis on cases where retrieval fails and where overall task fails. This section illustrates representative examples of retrieved content, how R-WoM behaves when facing retrieval failure, and identifies major failure causes. Table 9 shows two representative examples that demonstrate how retrieval behaves in distinct scenarios. When the user query contains explicit action semantics, the module successfully retrieves relevant procedural content. In contrast, failures tend to occur when the task requires implicit reasoning or semantic inference, which leads to ambiguous or misleading retrievals. These examples show that retrieval succeeds when task descriptions explicitly describe the intended operation but fails when the request requires contextual understanding or multi-step reasoning (e.g., manipulation of some specific data). In such cases, the textual query may not contain enough cues for the retriever to locate a matching tutorial.

We further analyze the end-to-end performance when retrieval failure happens and the failure-related statistics can be found in Table 10. From the statistics, we observe that the R-WoM behaves consistently as WebDreamer in all of the cases when retrieval fails. Specifically, in tasks where using world model without grounding can succeed, adding retrieved content will not lead to task failure even if the retrieved content is not directly relevant. In some cases, R-WoM still completes the task successfully even when the retrieved tutorial is irrelevant. We conjecture the reason behind this phenomenon is heuristic filtering strategies we use as below:

- **Reranker-level filtering.** The LLM-based reranker is explicitly prompted to judge semantic relevance rather than relying purely on embedding cosine similarity. This allows it to discard tutorials that are lexically similar but semantically unrelated to the target goal.

- **Rollout-level filtering.** During world-model rollouts, the policy is instructed to incorporate retrieved content only when it aligns with the current observation (layout, UI elements, and goal). If the tutorial is mismatched, the model falls back on internal reasoning instead of being influenced by irrelevant information.

Moreover, we also investigate the failure cases of R-WoM, and Table 11 summarizes the statistics of failures across different task types. Specifically, following the task taxonomy used in the WMA

Table 11: Failure distribution by task type across OSWorld and WebArena.

| Benchmark | Model | Information-Seeking / Navigation | Content-Modification |
|---|---|---|---|
| OSWorld | Qwen2.5-VL-72B | 23.5% | 76.5% |
| | Claude-3.5-Sonnet | 21.8% | 78.2% |
| | Claude-3.7-Sonnet | 19.6% | 80.4% |
| WebArena | Qwen2.5-VL-72B | 37.4% | 62.6% |
| | Claude-3.5-Sonnet | 35.7% | 64.3% |
| | Claude-3.7-Sonnet | 34.8% | 65.2% |

Table 12: Full-procedure alignment scores given by different LLM judges.

| Model | Claude-3.7-Sonnet | GPT-4.1 | Gemini-2.5-Pro |
|---|---|---|---|
| Qwen2.5-VL-72B-Instruct | 50.0% | 50.0% | 45.0% |
| Claude-3.5-Sonnet | 55.0% | 55.0% | 50.0% |
| Claude-3.7-Sonnet | 65.0% | 60.0% | 65.0% |

Table 13: Performance comparison of different methods on stronger model backbones (Claude-4-Sonnet and Claude-4.5-Sonnet) evaluated on the OSWorld benchmark.

| Model | Claude-4-Sonnet | Claude-4.5-Sonnet |
|---|---|---|
| Vanilla | 48.24% | 59.12% |
| RAG | 50.82% | 60.43% |
| WoM | 49.65% | 62.09% |
| R-WoM | 56.73% | 67.84% |

(Chae et al., 2024), we categorize tasks into two major groups: (i) information-seeking or navigation, and (ii) content-modification (i.e., editing or manipulating the content of a website, file, or software). From these results, we observe that content-modification tasks (e.g., editing text in LibreOffice or modifying code in VSCode) are where R-WoM fails more frequently. These failures often arise from ambiguous task goals (e.g., "Please calculate the ages of the employees according to their birthday"), where the model must infer intermediate steps, or from the inherent difficulty of manipulating fine-grained content through UI actions such as dragging, selecting, or highlighting text. Such tasks remain challenging for current computer-use agents because they require both precise visual grounding and multi-step reasoning. Exploring richer strategies such as multi-modal retrieval or more agentic, step-aware retrieval mechanisms represents a promising direction for future improvement.

## A.6 MORE ABLATION STUDIES

**Using different LLMs as judges for full-procedure planning alignment.** To verify that our evaluation is not biased toward a specific judge model, we conduct a cross-judge consistency ablation. Specifically, we use Claude-3.7-Sonnet, GPT-4.1, and Gemini-2.5-Pro, on the same set of tasks mentioned in Section 3.2. These results show that alignment judgments remain broadly consistent across different judge models, indicating that our evaluation is not overly dependent on a single model family and is robust to variations in judgment sources.

**End-to-end performance when using more powerful LLMs.** To examine whether the proposed R-WoM generalizes effectively across stronger model backbones, we extend our evaluation to two more up-to-date models: Claude-4-Sonnet and Claude-4.5-Sonnet, both of which have been optimized for computer-use environments. The results are presented in Table 13. We evaluate them on the OSWorld benchmark following the same experimental setup described in Section 5.1 of our paper. Across both stronger models, we observe consistent performance improvements after integrating R-WoM. These results confirm that the proposed framework continues to deliver substantial gains even on more capable language models.

Table 14: Comparison between R-WoM (listwise reward) and its absolute reward variant on the OSWorld benchmark.

| Model | R-WoM | R-WoM (Absolute Reward Variant) |
|---|---|---|
| Qwen2.5-VL-72B-Instruct | 38.05% | 35.12% |
| Claude-3.5-Sonnet | 26.41% | 24.03% |
| Claude-3.7-Sonnet | 39.13% | 36.50% |

Table 15: Ablation on the number of candidate actions ($m$) for listwise reward optimization, evaluated on the OSWorld benchmark.

| Action Candidate Size ($m$) | Claude-3.7-Sonnet | Claude-4-Sonnet | Claude-4.5-Sonnet |
|---|---|---|---|
| 2 | 37.90% | 53.67% | 66.00% |
| 3 | 39.13% | 56.73% | 67.84% |
| 5 | 38.10% | 56.22% | 68.41% |

**The impact of listwise relative reward and sample-wise absolute reward.** To analyze the effectiveness of the listwise reward design in R-WoM, we conduct an ablation comparing our listwise reward design with an absolute reward variant. The absolute reward variant is implemented by following WebDreamer and WebEvolver to use $\{0, 0.5, 1.0\}$ to represent failure, on-the-progress and success, respectively. As illustrated in Table 14, the listwise reward consistently outperforms the absolute reward variant, confirming that relative ranking among candidates provides stronger learning signals and leads to more robust action selection during rollout.

**The impact of candidate size on listwise reward.** To further study the impact of candidate action set on the effectiveness of listwise reward optimization, we conduct an ablation study by varying the number of candidate actions $m$ and Table 15 reports the performance when varying the candidate size $m = \{2, 3, 5\}$. The results show that increasing the candidate size from 2 to 3 improves overall performance across all models, suggesting that moderate expansion of the comparison set helps the judge LLM better identify globally optimal actions. However, further increasing the size to 5 can have diminishing returns when the base model is relatively weak (e.g., Claude-3.7-Sonnet and Claude-4-Sonnet).

A.7 USE OF LARGE LANGUAGE MODELS

We utilized Large Language Models (LLMs), such as Claude, exclusively for ancillary support in two main areas: (i) language editing and polishing of the manuscript, and (ii) coding assistance for minor boilerplate tasks, such as generating plotting scripts and small utilities. All model-generated outputs were thoroughly reviewed, modified, and rigorously tested by the authors to ensure their accuracy and appropriateness.

