# OpenReview forum: "R-WoM: Retrieval-augmented World Model For Computer-use Agents"
_ICLR.cc/2026/Conference — ICLR 2026 Poster_

### Official Review · Reviewer_1XKm · 2025-10-24

**Soundness:** 2
**Presentation:** 2
**Contribution:** 3
**Rating:** 6
**Confidence:** 3

**Summary:**

This paper proposes R-WoM (Retrieval-augmented World Model), a framework designed to enhance computer-use agents. Traditional world models for agents often struggle with long-horizon reasoning. R-WoM addresses the issue by integrating retrieval-augmented memory into a world model.
The method is evaluated on benchmarks of computer-use agents, showing that R-WoM significantly outperforms baseline world models and state-of-the-art methods in terms of task success rate, with particular advantage in longer-horizon simulations.

Overall, this is a strong and well-motivated paper. It contributes a novel and effective combination of retrieval and world models, demonstrating strong empirical results on computer-use agents. Its main areas for improvement are in experiment soundness, experimental analysis breadth & deepness, scalability, and retrieval interpretability. With these addressed, the work could be highly impactful for the development of practical, general-purpose computer-use assistants.

**Strengths:**

- This paper addresses an increasingly important challenge: enabling LLM-powered agents to use computers effectively.
- Novel integration of retrieval augmentation with world models in the specific context of computer-use agents.
- This paper demonstrates that retrieval + world models can substantially improve generalization and robustness. Experimental evaluation shows consistent improvements over baselines across two comprehensive benchmarks designed for multi-round interactions in realistic computer-use environments.
- This paper is well-written and logically structured, in spite of more experimental analysis are advised.
- Potentially impactful for building personal assistant agents capable of robust computer-use in real-world settings.

**Weaknesses:**

- Two benchmarks are not sound enough. Experiments are limited to a specific set of tasks. It is unclear how R-WoM would perform on diverse real-world applications (e.g., enterprise software, web navigation with dynamic layouts).
- The evaluation lacks stress tests for extreme long-horizon/complex tasks (hundreds of steps).
- This paper does not deeply analyze the computational overhead of retrieval at inference time. Scaling to large memory banks or diverse user contexts could introduce latency.
- While retrieval helps generalization, it is not always clear what is retrieved and why. More analysis of retrieval quality and relevance would strengthen the contribution.
- The world model’s predictive accuracy is not reported in detail (e.g., prediction error across horizons). Understanding when it fails would provide useful insights.
- While strong baselines are included, it would be useful to see how R-WoM compares with end-to-end planning agents trained with reinforcement learning at scale.

**Questions:**

- How well does R-WoM scale when retrieval memory grows to millions of trajectories? Do you employ approximate retrieval or pruning strategies?
- Have you tested R-WoM on completely unseen applications (e.g., apps with novel layouts or interaction paradigms)? How robust is the method in such settings?
- Can you provide examples of what the retrieval module retrieves in successful vs. failed cases? Does retrieval sometimes introduce misleading prior experiences?
- How accurate is the predictive component of the world model across multiple horizons? Do errors accumulate in long trajectories?
- In a real-world assistant scenario, how would you balance retrieval latency, model complexity, and user responsiveness?
- Could this framework generalize to other domains requiring long-horizon reasoning (e.g., robotics, scientific workflows), or is it specialized to computer-use tasks?

---

> ### Author Response · Authors · 2025-11-22
> **Responses to Reviewer 1XKm (Part 1/4)**
>
> > ***W1: Two benchmarks are not sound enough. Experiments are limited to a specific set of tasks. It is unclear how R-WoM would perform on diverse real-world applications (e.g., enterprise software, web navigation with dynamic layouts).***
>
> > ***W2: The evaluation lacks stress tests for extreme long-horizon/complex tasks (hundreds of steps).***
>
> We understand your concern regarding the breadth of our evaluation. We would like to clarify that the two benchmarks we employ, OSWorld and WebArena, are not only widely used in academic research but are also adopted directly in the official model evaluations of top-tier frontier LLMs, including Qwen [1], GPT [2] and Claude [3], and others. These model families use OSWorld and WebArena as part of their agent-evaluation.
>
> Beyond coverage, these benchmarks already contain complex environments including multi-step software workflows, cross-application operations, dynamic UI variations. Several recent works [4,5] conduct online evaluations using only one of these benchmarks, underscoring that each alone is already considered sufficiently challenging.
>
> Regarding the reviewer’s concern about long-horizon complexity (hundreds of steps), we note that existing benchmarks rarely include such extremely long trajectories because even single-action grounding remains challenging for current LLM-based agents. However, to examine the potential of R-WoM in longer-horizon tasks, we conducted further analysis of our results by step ranges as below:
>
> **OSWorld step-range breakdown**
>
> | Model             | Method  | Success ≤25 steps | Success >25 steps |
> | ----------------- | ------- | ----------------- | ----------------- |
> | Qwen-2.5-VL-72B-Instruct   | Vanilla | 16                | 6                 |
> |                   | RAG     | 18                | 8                 |
> |                   | WoM     | 17                | 7                 |
> |                   | R-WoM   | 21                | 11                |
> | Claude-3.5-Sonnet | Vanilla | 14                | 5                 |
> |                   | RAG     | 13                | 6                 |
> |                   | WoM     | 15                | 5                 |
> |                   | R-WoM   | 15                | 7                 |
> | Claude-3.7-Sonnet | Vanilla | 18                | 6                 |
> |                   | RAG     | 17                | 7                 |
> |                   | WoM     | 19                | 7                 |
> |                   | R-WoM   | 22                | 11                |
>
> **WebArena step-range breakdown**
>
> | Model             | Method  | Success ≤15 steps | Success >15 steps |
> | ----------------- | ------- | ----------------- | ----------------- |
> | Qwen-2.5-VL-72B-Instruct   | Vanilla | 16                | 8                 |
> |                   | RAG     | 17                | 10                |
> |                   | WoM     | 18                | 12                |
> |                   | R-WoM   | 21                | 12                |
> | Claude-3.5-Sonnet | Vanilla | 18                | 10                |
> |                   | RAG     | 19                | 12                |
> |                   | WoM     | 20                | 13                |
> |                   | R-WoM   | 20                | 17                |
> | Claude-3.7-Sonnet | Vanilla | 20                | 12                |
> |                   | RAG     | 21                | 15                |
> |                   | WoM     | 22                | 16                |
> |                   | R-WoM   | 23                | 17                |
>
> The results show that a substantial portion of R-WoM’s gain appears in longer-step trajectories even though the base success rate there is lower, suggesting the potential of our framework to more complex long-horizon scenarios. Besides, we view expanding to enterprise software or extremely long sequences as promising and important future directions and are actively exploring these extensions.

---

> > ### Author Response · Authors · 2025-11-22
> > **Responses to Reviewer 1XKm (Part 2/4)**
> >
> > > ***W3: This paper does not deeply analyze the computational overhead of retrieval at inference time.***
> >
> > > ***Q1: How well does R-WoM scale when retrieval memory grows to millions of trajectories? Do you employ approximate retrieval or pruning strategies?***
> >
> > To clarify the computational overhead of retrieval, we first emphasize that R-WoM retrieves tutorials rather than full trajectories. Unlike trajectories, which can easily scale to millions, official tutorials are highly curated, abstract, and general-purpose, so the total number of available tutorials is far smaller and does not approach million-level scale. Besides, modern RAG systems already rely on approximate nearest neighbor (ANN) search (e.g., FAISS, ScaNN), which is the standard solution for large-scale retrieval and scales well to millions of indexed documents.
> >
> > In addition, our framework incorporates two pruning mechanisms to further reduce retrieval cost and improve relevance: (1) using domain names as metadata to restrict the search space, and (2) applying an LLM-based semantic filter during reranking to remove irrelevant tutorials. To further contextualize the relative overhead, the table below reports the average retrieval time per sample (including the query rewriting and reranking) and the average total end-to-end execution time per task sample on the OSWorld benchmark:
> >
> > | Model          | Avg time of retrieval per task sample | Avg time of overall execution per task sample |
> > | -------------- | -------------------------------- | ---------------------------------------- |
> > | Qwen2.5-VL-72B-Instruct | 5.1s                             | 989s                                     |
> > | Claude-3.5-Sonnet      | 5.3s                             | 518s                                     |
> > | Claude-3.7-Sonnet      | 4.7s                             | 410s
> >
> > These measurements show that retrieval accounts for less than 2 percent of total execution time across models. Therefore, even if the retrieval index grows substantially, the overhead remains minor relative to the full execution pipeline.

---

> > > ### Author Response · Authors · 2025-11-22
> > > **Responses to Reviewer 1XKm (Part 3/4)**
> > >
> > > > ***W4: While retrieval helps generalization, it is not always clear what is retrieved and why.***
> > >
> > > > ***Q3: Can you provide examples of what the retrieval module retrieves in successful vs. failed cases? Does retrieval sometimes introduce misleading prior experiences?***
> > >
> > > To illustrate what the retrieval module returns in different scenarios, we provide concrete examples of both successful and failed retrievals:
> > >
> > > | Case Type  | User Query                                                                                        | Retrieved Tutorial                                                                                                                                                                   |
> > > | ---------- | ------------------------------------------------------------------------------------------------- | ------------------------------------------------------------------------------------------------------------------------------------------------------------------------------------ |
> > > | Successful | “Computer, can you turn the webpage I'm looking at into a PDF file and put it on my main screen?” | “Print from Chrome — On your computer, open Chrome. Open the page, image, or file you want to print. Click File → Print, or use Ctrl+P / ⌘+P. Select ‘Save as PDF’ and click Print...” |
> > > | Failure    | “Please calculate the ages of the employees according to their birthday.”                         | “When entering a time, separate time elements with colons. To change the date or time format in Calc, open Format Cells and select Date/Time...”                                   |
> > >
> > > As the examples show, retrieval can return highly relevant content especially when the task description contains clear action semantics. However, failures primarily occur in tasks that require semantic inference or multi-step reasoning, such as content modification in documents or spreadsheets. These failures often arise because the textual query is ambiguous, making it difficult for retrieval to identify the correct instructional tutorial.
> > >
> > > To mitigate the effect of irrelevant tutorials, we have already applied two forms of heuristic filtering as below:
> > >
> > > 1. Reranker-level filtering. When using the LLM reranker, we explicitly prompt the model to judge semantic relevance instead of relying solely on cosine similarity. This allows it to filter out tutorials that rank high in embedding space but are not actually useful for the task.
> > > 2. Rollout-level filtering. During world-model rollouts, we instruct the LLM to incorporate tutorial content only when it matches the current observation (layout, UI elements, stated goal). If the retrieved document appears irrelevant or mismatched, the model is guided to ignore it and rely on its own internal reasoning.
> > >
> > > Below we summarize retrieval-related statistics for both OSWorld and WebArena:
> > >
> > > **Statistics on OSWorld**
> > >
> > > | Metric                                                   | Qwen2.5-VL-72B-Instruct | Claude-3.5-Sonnet | Claude-3.7-Sonnet |
> > > | -------------------------------------------------------- | -------------- | ----------------- | ----------------- |
> > > | Retrieval failures                                       | 5              | 4                 | 4                 |
> > > | Successful tasks despite retrieval failure               | 3              | 3                 | 3                 |
> > > | Failures shared with WebDreamer (i.e., not caused by retrieval) | 2              | 1                 | 1                 |
> > >
> > > **Statistics on WebArena**
> > >
> > > | Metric                                                   | Qwen2.5-VL-72B | Claude-3.5-Sonnet | Claude-3.7-Sonnet |
> > > | -------------------------------------------------------- | -------------- | ----------------- | ----------------- |
> > > | Retrieval failures                                       | 17             | 17                | 16                |
> > > | Successful tasks despite retrieval failure               | 11             | 12                | 11                |
> > > | Failures shared with WebDreamer (i.e., not caused by retrieval) | 6              | 5                 | 5                 |
> > >
> > > In the rare cases where retrieval fails, the world model continues to condition on the current screen state and full action history, allowing it to fall back on its internal reasoning without being misled by irrelevant information. This helps explain why the R-WoM can have comparable behaviors as WebDreamer when retrieval fails.

---

> ### Author Response · Authors · 2025-11-22
> **Responses to Reviewer 1XKm (Part 4/4)**
>
> > ***W5: The world model’s predictive accuracy is not reported in detail (e.g., prediction error across horizons).***
>
> > ***Q4: How accurate is the predictive component of the world model across multiple horizons? Do errors accumulate in long trajectories?***
>
> We acknowledge that evaluating world-model rollout accuracy at each individual step is inherently difficult, as it requires fine-grained labeling of predicted intermediate UI states and actions, which is extremely costly to annotate at scale. Nonetheless, we have already conducted analysis on how the world model behaves under different imagination horizons.
>
> As we have shown in Figure 4 of our paper, varying the imagination horizon provides an indirect but meaningful view of how prediction errors accumulate during multi-step rollouts. We observe that, for the base WoM models, performance often degrades at longer horizons, which suggests that hallucinations and compounding errors emerge as imagination depth increases. With retrieval augmentation, however, R-WoM sustains higher accuracy across horizons and reaches peak performance at a larger imagination depth (around horizon 3). This indicates that grounding with retrieved task-relevant knowledge reduces accumulated errors and helps stabilize long-horizon prediction.
>
> However, once the horizon becomes too long, performance still declines even for R-WoM. This implies that improving stability over very long imagination rollouts remains an important direction for future work.
>
>
> > ***Q2: Have you tested R-WoM on completely unseen applications (e.g., apps with novel layouts or interaction paradigms)? How robust is the method in such settings?***
>
> > ***Q6: Could this framework generalize to other domains requiring long-horizon reasoning (e.g., robotics, scientific workflows), or is it specialized to computer-use tasks?***
>
> To evaluate how R-WoM handles scenarios beyond the environments seen during training, we first note that constructing truly unseen applications with new UIs or layouts is non-trivial as altering interfaces in OSWorld and WebArena are time-consuming. Nevertheless, to approximate this challenge, we assess R-WoM in a tutorial-scarce setting within tasks in the OSWorld for which online documentation is difficult to find and therefore might be less likely to appear in LLM pretraining corpora. In these cases, we synthesize tutorials automatically from OpenCUA [4]'s collected trajectories via self-play, and then apply R-WoM using these synthetic tutorials. The performance results across three models are shown below.
>
> | Model      | Claude-3.7-Sonnet | Claude-4-Sonnet | Claude-4.5-Sonnet |
> | ---------- | ---------- | -------- | ---------- |
> | Vanilla    | 32.25%     | 35.82%   | 45.83%     |
> | RAG        | 33.36%     | 36.93%   | 46.11%     |
> | WebDreamer | 30.86%     | 34.43%   | 46.35%     |
> | R-WoM      | 35.71%     | 39.28%   | 49.29%     |
>
> The consistent improvements across all models indicate that even when no high-quality human-written tutorials exist, R-WoM can still operate effectively by synthesizing tutorials from trajectories. This demonstrates the method’s ability to extend to settings where tasks might be less likely to have been seen during pre-training.
>
> Beyond OSWorld, we also expect R-WoM to generalize well to other domains where actions are irreversible or where workflows are long-horizon and structured. Domains such as robotics, laboratory automation, and scientific computing often provide rich manuals, procedural documentation, and step-by-step references—making them naturally compatible with retrieval-augmented grounding. Studies have already shown that these environments could benefit from retrieval-augmentation [5]. Therefore, R-WoM is also very likely to well-documented instructions to reduce errors and improve long-horizon reasoning. Exploring such applications is a promising direction for future work.
>
> > ***Q5: In a real-world assistant scenario, how would you balance retrieval latency, model complexity, and user responsiveness?***
>
> Please refer to our discussion in the general response.
>
> [1] Qwen3 model card: https://huggingface.co/Qwen/Qwen3-VL-8B-Instruct
>
> [2] GPT computer-use model card: https://openai.com/index/computer-using-agent/
>
> [3] Claude4.5 model card: https://www.anthropic.com/news/claude-sonnet-4-5
>
> [4] Wang et al. OpenCUA: Open Foundations for Computer-Use Agents. NeurIPS 2025
>
> [5] Zhu et al. Retrieval-Augmented Embodied Agents. CVPR 2024

---

> ### Comment · Reviewer_1XKm · 2025-11-26
> **Response to Authors' Rebuttal**
>
> Dear Authors,
>
> Thank you very much for the clarification! Most of them have addressed my concerns.
>
> I appreciate the response, especially the newly provided statistics, cases and experiments. Including those in the revised version will strengthen the paper.
>
> Overall, I acknowledge the general quality this work, but still have reservations about:
> - In your rebuttal to W1 & W2, I find that the total quantities of cases for each model aren't equal. For example, Claude-3.7-Sonnet, Vanilla: 18 + 6 = 24, while R-WoM: 22 + 11 = 33, how do you claim that ''a substantial portion of R-WoM’s gain appears in longer-step trajectories even though the base success rate there is lower'', I don't think the comparion is fare enough.
> - Q2 and Q6 are not the same...
>
>
> Best Regards,
>
> Reviewer 1XKm

---

> ### Author Response · Authors · 2025-11-26
>
> Dear reviewer,
>
> Thank you for your recognition of the quality of our work and we have updated the new results in our revised paper. And we would like to make further clarifications on your remaining concerns:
>
> 1. To better address W1 and W2, we compare the improvements of R-WoM over other baselines on tasks with different trajectory lengths (e.g., steps ≤ 25 vs. steps > 25 on OSWorld). Note that and the number of steps are calculated over the steps on the tasks where agents can complete the tasks successfully. This is because the dataset itself does not have a ground truth step for achieving each task so a more practical way is to calculate the steps that agents actually take to achieve the tasks. And for the improvement analysis, for instance, on OSWorld under claude-3.7-sonnet, R-WoM outperforms the second-best baseline by 3 points (22 – 19) on completed tasks with ≤ 25 steps, and by 4 points (11 – 7) on completed tasks with > 25 steps. This demonstrates that R-WoM can have the potential of achieving consistent, and even stronger, improvements on task completion requiring longer steps.
>
> 2. We apologize for the earlier confusion. The typos have been corrected in the revised version of our responses.
>
> Best,

---

### Official Review · Reviewer_tkQf · 2025-10-28

**Soundness:** 2
**Presentation:** 2
**Contribution:** 2
**Rating:** 4
**Confidence:** 3

**Summary:**

This paper investigates the limitations of using Large Language Models as world models for computer-use agents, particularly their unreliability in long-horizon planning due to hallucinations and static knowledge. To diagnose this, the authors first conduct a systematic probing analysis, revealing that while LLMs excel at short-term state prediction and recognizing meaningful progress, they fail to generate plans that align with real environment dynamics over extended steps. To address this, the paper proposes the Retrieval-augmented World Model, a framework that grounds the LLM's simulation process in external knowledge retrieved from tutorials. Key components of R-WoM include a reasoning-based RAG pipeline for better retrieval, a CoT-based mechanism for efficient multi-step rollouts, and a listwise reward estimation strategy for more robust action selection. Experiments on two challenging benchmarks, OSWorld and WebArena, show that R-WoM significantly outperforms strong baselines across multiple LLM backbones.

**Strengths:**

Clear Motivation: The paper is motivated by a systematic probing analysis that pinpoints the specific failure modes of ungrounded LLM-based world models in long-horizon tasks, providing a clear foundation for the research.

Reasonable Framework Design: R-WoM is a reasonably designed framework that combines ideas like a RAG pipeline, CoT-based simulation, and listwise reward estimation.

Evaluation on Multiple Benchmarks: The evaluation demonstrates performance gains over baselines across two challenging benchmarks and three different LLMs.

**Weaknesses:**

Limited Novelty: The core idea of the paper is to apply Retrieval-Augmented Generation (RAG) to world models. While this is a logical engineering direction, it feels more like a direct combination of existing, mature technologies (RAG, CoT, World Models) and lacks fundamental theoretical or methodological innovation. This makes the paper's contribution feel more applied and engineering-focused rather than a breakthrough in basic research.

Severe Generalization Limits: The method's effectiveness is severely constrained to domains that have high-quality, comprehensive tutorials. This is a critical flaw, as it means the method is nearly inapplicable in many real-world scenarios where documentation is sparse, outdated, or non-existent. The paper fails to adequately discuss or experimentally verify this limitation, which significantly undermines its claims of general utility.

Unfavorable Cost-Benefit Ratio: Although R-WoM is more efficient than the WebDreamer baseline, its computational overhead is still excessive compared to a standard RAG approach. The results show that achieving a limited performance gain (sometimes in the single digits) requires a several-fold increase in computational cost. This cost-benefit trade-off is unacceptable for many practical applications and casts doubt on the method's utility.

Effectiveness of Core Component is Questionable: The results with an "oracle" retriever expose the performance bottleneck of the current retrieval module. This is not just a technical detail; it directly challenges the paper's central thesis that world models can be effectively grounded via retrieval. If the retrieval itself is unreliable, the foundation of the entire framework is shaky.

**Questions:**

1. Could you comment on the expected performance of R-WoM in environments with limited or no available tutorial documentation? How gracefully does the model's performance degrade as the quality of the knowledge base decreases?

2. The listwise reward estimation is an interesting and intuitive design choice. Have you performed an ablation study to specifically isolate its contribution? For instance, how does the full R-WoM framework compare to a version that uses a more traditional absolute reward scoring (e.g., a 1-10 scale) on the generated rollouts?

3. Table 5 shows a clear trade-off between cost and performance. For example, using Claude-3.7 on WebArena, R-WoM is ~4.75x slower than RAG for a ~7.2% relative improvement. In what practical scenarios do you believe this trade-off is justified?

---

> ### Author Response · Authors · 2025-11-22
> **Responses to Reviewer tkQf (Part 1/3)**
>
> > ***W1: Limited Novelty: The core idea of the paper is to apply Retrieval-Augmented Generation (RAG) to world models. While this is a logical engineering direction, it feels more like a direct combination of existing, mature technologies (RAG, CoT, World Models) and lacks fundamental theoretical or methodological innovation. This makes the paper's contribution feel more applied and engineering-focused rather than a breakthrough in basic research.***
>
> While our framework builds on established components such as RAG and world-model rollouts, our contribution is not a simple combination of existing techniques. First, our probing study identifies a fundamental weakness in current LLM-based world models in realistic environments: their limitations in environment-specific reasoning due to LLMs' stale or incomplete parametric knowledge. R-WoM directly targets this bottleneck by introducing retrieval-augmented grounding that dynamically injects up-to-date procedural knowledge into the world-modeling process. Second, our work provides a comprehensive empirical analysis across two challenging computer-use benchmarks, revealing when and how retrieval helps, and how world-model rollouts behave under grounded vs. ungrounded conditions. This type of empirical contribution is well aligned with prior empirical studies such as [1,2,3,4], which similarly advance the field by exposing limitations of current LLM agents and demonstrating practical mechanisms to overcome them.
>
> > ***W2: Severe Generalization Limits: The method's effectiveness is severely constrained to domains that have high-quality, comprehensive tutorials. This is a critical flaw, as it means the method is nearly inapplicable in many real-world scenarios where documentation is sparse, outdated, or non-existent. The paper fails to adequately discuss or experimentally verify this limitation, which significantly undermines its claims of general utility.***
>
> > ***Q1: Could you comment on the expected performance of R-WoM in environments with limited or no available tutorial documentation? How gracefully does the model's performance degrade as the quality of the knowledge base decreases?***
>
> We would like to highlight that the core mechanism of R-WoM is to leverage environment-specific knowledge to guide the world model in imagining the consequences of actions, thereby improving decision-making. Even in domains where high-quality tutorials are scarce, R-WoM can still be applied by synthesizing tutorials from self-played agent trajectories. This allows the world model to learn task-relevant procedural knowledge even when human-written tutorials are limited.
>
> To demonstrate this capability, we extend R-WoM to the tutorial-scarce OSWorld setting by synthesizing tutorials using the self-played trajectories provided in OpenCUA [5]. We then do the comparison against baselines using three models: Claude-3.7-Sonnet, Claude-4-Sonnet, and Claude-4.5-Sonnet. The performance comparison is shown below.
>
> | Model   | Claude-3.7-Sonnet | Claude-4-Sonnet | Claude-4.5-Sonnet |
> |---------|------------|-----------|-------------|
> | Vanilla | 32.25%     | 35.82%    | 45.83%      |
> | RAG     | 33.36%     | 36.93%    | 46.11%      |
> | WebDreamer     | 30.86%     | 34.43%    | 46.35%      |
> | R-WoM    | 35.71%     | 39.28%    | 49.29%      |
>
> The consistent improvement across the three models indicates that even in tutorial-scarce scenarios, R-WoM can still maintain applicable.
>
> > ***W3: Unfavorable Cost-Benefit Ratio: Although R-WoM is more efficient than the WebDreamer baseline, its computational overhead is still excessive compared to a standard RAG approach. The results show that achieving a limited performance gain (sometimes in the single digits) requires a several-fold increase in computational cost. This cost-benefit trade-off is unacceptable for many practical applications and casts doubt on the method's utility.***
>
> > ***Q3: Table 5 shows a clear trade-off between cost and performance. For example, using Claude-3.7 on WebArena, R-WoM is ~4.75x slower than RAG for a ~7.2% relative improvement. In what practical scenarios do you believe this trade-off is justified?***
>
> Please refer to our response in the general response.

---

> > ### Author Response · Authors · 2025-11-22
> > **Responses to Reviewer tkQf (Part 2/3)**
> >
> > > ***W4: Effectiveness of Core Component is Questionable: The results with an "oracle" retriever expose the performance bottleneck of the current retrieval module. This is not just a technical detail; it directly challenges the paper's central thesis that world models can be effectively grounded via retrieval. If the retrieval itself is unreliable, the foundation of the entire framework is shaky.***
> >
> > We would like to acknowledge that retrieval can have failures but we would like to argue that the retrieval failure does not always lead to task achievement failure. One aspect is that the failure of retrieval just introduces irrelevant information rather than misinformation since our source of tutorials are mostly from official online websites, which are mostly reliable. Therefore, the retrieved knowledge is most likely to be irrelevant rather than incorrect.
> > Another aspect is that in our framework, we have already applied two forms of heuristic filtering to mitigate the effects of irrelevant tutorials, which are:
> >
> > 1. Reranker-level filtering. When using the LLM reranker, we explicitly prompt the model to judge semantic relevance instead of relying solely on cosine similarity. This allows it to filter out tutorials that rank high in embedding space but are not actually useful for the task.
> > 2. Rollout-level filtering. During world-model rollouts, we instruct the LLM to incorporate tutorial content only when it matches the current observation (layout, UI elements, stated goal). If the retrieved document appears irrelevant or mismatched, the model is guided to ignore it and rely on its own internal reasoning.
> >
> > The tables below summarize these statistics regarding detailed analysis of task achievement related to retrieval failure in R-WoM:
> >
> > **Statistics on OSWorld**
> >
> > | Metric                                                   | Qwen2.5-VL-72B-Instruct | Claude-3.5-Sonnet | Claude-3.7-Sonnet |
> > | -------------------------------------------------------- | -------------- | ----------------- | ----------------- |
> > | Retrieval failures                                       | 5              | 4                 | 4                 |
> > | Successful tasks despite retrieval failure               | 3              | 3                 | 3                 |
> > | Failures shared with WebDreamer (i.e., not caused by retrieval) | 2              | 1                 | 1                 |
> >
> > **Statistics on WebArena**
> >
> > | Metric                                                   | Qwen2.5-VL-72B-Instruct | Claude-3.5-Sonnet | Claude-3.7-Sonnet |
> > | -------------------------------------------------------- | -------------- | ----------------- | ----------------- |
> > | Retrieval failures                                       | 17             | 17                | 16                |
> > | Successful tasks despite retrieval failure               | 11             | 12                | 11                |
> > | Failures shared with WebDreamer (i.e., not caused by retrieval) | 6              | 5                 | 5                 |
> >
> > These results show that even when retrieval returns suboptimal content, the agent can still complete some tasks, and the remaining failures overlap heavily with the baseline WoM rather than being caused by irrelevant tutorials. This result suggests that the retrieval failure in R-WoM do not bring much additional misguidance on the agents, indicating the robustness of our framework.

---

> > > ### Author Response · Authors · 2025-11-22
> > > **Responses to Reviewer tkQf (Part 3/3)**
> > >
> > > > ***Q2: The listwise reward estimation is an interesting and intuitive design choice. Have you performed an ablation study to specifically isolate its contribution? For instance, how does the full R-WoM framework compare to a version that uses a more traditional absolute reward scoring (e.g., a 1-10 scale) on the generated rollouts?***
> > >
> > > To show the effectiveness of listwise reward, we do the ablation study on the reward function. Specifically, we adopt the discrete rewards {0, 0.5, 1.0} to suggest failure, on-the-progress and success, the same as WebDreamer and WebEvolver did. Here are their results on the OSWorld.
> > >
> > > | Model             | R-WoM  | R-WoM (Absolute Reward Variant) |
> > > | ----------------- | ------ | ------------------------------- |
> > > | Qwen-2.5-VL-72B-Instruct   | 38.05% | 35.12%                           |
> > > | Claude-3.5-Sonnet | 26.41% | 24.03%                           |
> > > | Claude-3.7-Sonnet | 39.13% | 36.50%                           |
> > >
> > > The results show that using an absolute reward scale can cause around 2-3% performance degrade. This suggests that when using absolute reward, it is difficult for LLMs to distinguish subtle differences between similar-quality rollouts, thus making the ordering less reliable.
> > >
> > > [1] Yao et al. ReAct: Synergizing Reasoning and Acting in Language Models. ICLR 2023
> > >
> > > [2] Wu et al. AutoGen: Enabling Next-Gen LLM Applications via Multi-Agent Conversation. COLM 2024
> > >
> > > [3] He et al. WebVoyager: Building an End-to-End Web Agent with Large Multimodal Models. ACL 2024
> > >
> > > [4] Gu et al. Is Your LLM Secretly a World Model of the Internet? Model-Based Planning for Web Agents. TMLR 2025
> > >
> > > [5] Wang et al. OpenCUA: Open Foundations for Computer-Use Agents. NeurIPS 2025

---

### Official Review · Reviewer_Cuhc · 2025-10-28

**Soundness:** 3
**Presentation:** 3
**Contribution:** 3
**Rating:** 6
**Confidence:** 4

**Summary:**

This paper investigates the capabilities and limitations of Large Language Models (LLMs) when used as "world models" for computer-use agents. The authors first conduct a systematic probing analysis across three tasks (next-state identification, full-procedure planning alignment, milestone transition recognition) to evaluate LLMs' ability to predict future states and estimate rewards. Their findings indicate that while LLMs excel at short-term predictions, their performance degrades significantly over longer horizons, failing to align with specific environment dynamics. To address this limitation, the paper introduces the R-WoM, a framework that grounds the LLM's simulation process in external, up-to-date knowledge retrieved from tutorials. R-WoM employs a reasoning-based retrieval pipeline and a listwise reward estimation strategy to improve the accuracy of multi-step rollouts. The authors demonstrate through experiments on the OSWorld and WebArena benchmarks that R-WoM substantially outperforms baseline methods, including vanilla agents, standard RAG, and an ungrounded world model (WebDreamer), particularly in tasks requiring long-horizon planning.

**Strengths:**

1.	Clear problem framing and probing suite. The three tasks isolate immediate state prediction vs. long-horizon planning vs. reward-related transition recognition, which yields diagnostic insights about LLM strengths/weaknesses.
2.	Effective and practical method. R-WoM is straightforward to implement (retrieval + rerank + CoT rollouts + listwise ranking) and integrates well with existing LLM stacks.
3.	Strong empirical gains. Consistent and sometimes large improvements across models and benchmarks (Table 2).
4.	Insightful ablations and visualization. Figures 3 provide clear comparisons of ungrounded, retrieval-based, and oracle-knowledge models. Figure 4 detail how imagination horizon impacts agent success, directly supporting claims about longer-horizon stability and providing actionable insights for the field.

**Weaknesses:**

1.	Limited novelty in core formulation. The high-level idea of retrieval-augmented LLMs for world modeling is not entirely new; prior works such as WorldGPT also consider augmentation or world modeling for digital environments. The contribution here is more in the detailed design (e.g., reasoning-based retrieval + listwise scoring) and empirical analysis than a fundamentally new methodological direction.
2.	Potential biases in evaluation. The use of LLM-based judges (Claude 3.5 Sonnet and Claude 3.7 Sonnet) to score plan alignment and trajectory success (especially in full-procedure planning) risks evaluation circularity (model judge vs. model being evaluated) and inherits bias from the same family of models. While necessary at present, some acknowledgment (and potentially a human evaluation subset) would bolster the claims.
3.	Selection bias toward tutorial-rich tasks. The method requires relevant tutorials; the authors explicitly sample task subsets that have tutorials available. The generalization to tutorial-scarce settings is unclear. Include experiments on a held-out set with sparse/no tutorials, or provide a quantitative analysis of how retrieval coverage correlates with gains.
4.	Limited Analysis of Failure Modes: While the paper demonstrates strong overall performance, a more comprehensive analysis of R-WoM's failure modes would strengthen the work. For instance, a discussion of scenarios where the model degrades (e.g., when faced with poor, irrelevant, or incorrect tutorials) or remains brittle (e.g., with ambiguous goals or novel UI elements) would provide significant value for practitioners seeking to build upon these findings.

**Questions:**

1.	Task subset selection: please quantify how many tasks were excluded due to absent tutorials and report the distribution of retrieval coverage (retrieval recall per task). This will help readers assess generality.  ￼
2.	Reranker details: Is it zero-shot or fine-tuned? How sensitive are results to the reranker model choice?
3.	Listwise reward computation: provide the exact mathematical definition (or pseudocode) of  in Eq. (11) $f_w(\{R(\hat \tau^{(j)},g,\mathcal{E})\})$ . How are rewards normalized across rollouts, and how does candidate set size m affect ranking quality?
4.	Judge bias: how consistent are the plan alignment judgments across different judge models or human annotators? Can you release a small human-annotated subset for reproducibility?
5.	Failure modes: can the authors provide qualitative examples where retrieval misleads the model (noisy / tangential tutorials) and explain any heuristic filters applied?

---

> ### Author Response · Authors · 2025-11-22
> **Responses to Reviewer Cuhc (Part 1/4)**
>
> > ***W1: Limited novelty in core formulation. The high-level idea of retrieval-augmented LLMs for world modeling is not entirely new; prior works such as WorldGPT also consider augmentation or world modeling for digital environments. The contribution here is more in the detailed design (e.g., reasoning-based retrieval + listwise scoring) and empirical analysis than a fundamentally new methodological direction.***
>
> Thank you for raising this important point. We agree that retrieval-augmented LLMs have been explored in prior work, such as WorldGPT. However, this earlier method does not introduce retrieval-augmented world modeling into multi-turn, realistic, procedural computer-use environments. WorldGPT focuses on improving multi-modal understanding such as video understanding and does not address the central challenges of GUI-based, action-diverse, multi-application workflows. This leaves a critical gap for exploring existing LLM as world models in the real computer-use settings. Moreover, our paper provides a comprehensive empirical investigation across two of the community’s challenging multi-turn environments (OSWorld and WebArena), offering actionable insights into when world model grounding helps and how retrieval stabilizes long-horizon predictions. This form of empirical advancement is well aligned with other empirical agent works such as ReAct [1], Autogen [2], Webvoyager [3] that advances the practical mechanisms.
>
> > ***W2: Potential biases in evaluation. The use of LLM-based judges (Claude 3.5 Sonnet and Claude 3.7 Sonnet) to score plan alignment and trajectory success (especially in full-procedure planning) risks evaluation circularity (model judge vs. model being evaluated) and inherits bias from the same family of models. While necessary at present, some acknowledgment (and potentially a human evaluation subset) would bolster the claims.***
>
> > ***Q4: Judge bias: how consistent are the plan alignment judgments across different judge models or human annotators? Can you release a small human-annotated subset for reproducibility?***
>
> Thank you for raising the concern about potential judge-model bias. To assess the robustness of our planning-alignment evaluation and reduce reliance on a single model family, we additionally employ two strong external LLM judges, GPT-4.1 and Gemini-2.5-Pro.
>
> | Model              | Claude-3.7-Sonnet | GPT-4.1 | Gemini-2.5-Pro |
> |--------------------|------------|---------|------------------|
> | Qwen-2.5-VL-72B-Instruct    | 50.0%      | 50.0%   | 45.0%            |
> | Claude-3.5-Sonnet  | 55.0%      | 55.0%   | 50.0%            |
> | Claude-3.7-Sonnet  | 65.0%      | 60.0%   | 65.0%            |
>
> These results show that alignment judgments remain broadly consistent across different judge models, suggesting that our evaluation is not overly dependent on a single model family.

---

> > ### Author Response · Authors · 2025-11-22
> > **Responses to Reviewer Cuhc (Part 2/4)**
> >
> > > ***W3: Selection bias toward tutorial-rich tasks. The method requires relevant tutorials; the authors explicitly sample task subsets that have tutorials available. The generalization to tutorial-scarce settings is unclear. Include experiments on a held-out set with sparse/no tutorials, or provide a quantitative analysis of how retrieval coverage correlates with gains.***
> >
> > > ***Q1: Task subset selection: please quantify how many tasks were excluded due to absent tutorials and report the distribution of retrieval coverage (retrieval recall per task). This will help readers assess generality.***
> >
> > First, we quantify tutorial availability across the benchmarks as below. In OSWorld, 85 tasks have clear and verifiable tutorial references from official online documents, while 276 tasks lack such references. For WebArena, based on its 301 unique task templates, 113 have tutorial coverage and 188 do not. This split is determined using a consistent criterion: whether a task’s goal can be matched to explicit operation instructions from official online tutorials or offline documentation for the target software or website.
> >
> > To address the generalization concern, we extend R-WoM to tasks where no online tutorials can be retrieved from the existing documents. We emphasize that even when external tutorials are scarce, R-WoM remains applicable by tutorial synthesis directly from self-played trajectories. This follows the same intuition as recent works that leverages procedural memory from self-play [4,5]. Specifically, we leverage 2k of the open-sourced trajectories released in OpenCUA [6] (note: these trajectories do not overlap with our test tasks) and adopt a two-stage synthesize–then–consolidate pipeline to obtain ~1.3k synthesized tutorials. These serve as general operation guidelines for tasks that lack online references. We then evaluate R-WoM using these synthesized tutorials with three models, and summarize the results below.
> >
> > | Model   | Claude-3.7-Sonnet | Claude-4-Sonnet | Claude-4.5-Sonnet |
> > | ------- | ---------- | -------- | ---------- |
> > | Vanilla | 32.25%     | 35.82%   | 45.83%     |
> > | RAG     | 33.36%     | 36.93%   | 46.11%     |
> > | WoM     | 30.86%     | 34.43%   | 46.35%     |
> > | R-WoM   | 35.71%     | 39.28%   | 49.29%     |
> >
> > The consistent improvement across all three models shows that R-WoM remains effective even when operating in tutorial-scarce settings. This demonstrates that R-WoM does not rely strictly on the existence of high-quality online manuals; instead, it can adapt to new tasks by grounding the world model using synthesized tutorials derived from **self-play**. Moreover, this design principle naturally generalizes to other domains with structured manuals, such as robotics and scientific workflows, where stepwise procedures are often irreversible and documentation is abundant, making retrieval-augmented grounding particularly beneficial.
> >
> > Overall, these results provide quantitative and qualitative evidence that the applicability of R-WoM extends beyond tutorial-rich tasks, and that tutorial coverage does not overly constrain its generalization.

---

> > > ### Author Response · Authors · 2025-11-22
> > > **Responses to Reviewer Cuhc (Part 3/4)**
> > >
> > > > ***W4: Limited Analysis of Failure Modes: While the paper demonstrates strong overall performance, a more comprehensive analysis of R-WoM's failure modes would strengthen the work. For instance, a discussion of scenarios where the model degrades (e.g., when faced with poor, irrelevant, or incorrect tutorials) or remains brittle (e.g., with ambiguous goals or novel UI elements) would provide significant value for practitioners seeking to build upon these findings.***
> > >
> > > > ***Q5: Failure modes: can the authors provide qualitative examples where retrieval misleads the model (noisy / tangential tutorials) and explain any heuristic filters applied?***
> > >
> > > Thank you for your suggestions. We have conducted a detailed analysis of task achievement related to retrieval failure in R-WoM. The tables below summarize these statistics:
> > >
> > > **Statistics on OSWorld**
> > >
> > > | Metric                                                   | Qwen2.5-VL-72B-Instruct | Claude-3.5-Sonnet | Claude-3.7-Sonnet |
> > > | -------------------------------------------------------- | -------------- | ----------------- | ----------------- |
> > > | Retrieval failures                                       | 5              | 4                 | 4                 |
> > > | Successful tasks despite retrieval failure               | 3              | 3                 | 3                 |
> > > | Failures shared with WoM (i.e., not caused by retrieval) | 2              | 1                 | 1                 |
> > >
> > > **Statistics on WebArena**
> > >
> > > | Metric                                                   | Qwen2.5-VL-72B-Instruct | Claude-3.5-Sonnet | Claude-3.7-Sonnet |
> > > | -------------------------------------------------------- | -------------- | ----------------- | ----------------- |
> > > | Retrieval failures                                       | 17             | 17                | 16                |
> > > | Successful tasks despite retrieval failure               | 11             | 12                | 11                |
> > > | Failures shared with WoM (i.e., not caused by retrieval) | 6              | 5                 | 5                 |
> > >
> > > These results show that even when retrieval returns suboptimal content, the agent can still complete some tasks, and the remaining failures overlap heavily with the baseline WoM rather than being caused by irrelevant tutorials. This is because we apply two forms of heuristic filtering to mitigate the effects of irrelevant tutorials:
> > >
> > > 1. Reranker-level filtering. When using the LLM reranker, we explicitly prompt the model to judge semantic relevance instead of relying solely on cosine similarity. This allows it to filter out tutorials that rank high in embedding space but are not actually useful for the task.
> > > 2. Rollout-level filtering. During world-model rollouts, we instruct the LLM to incorporate tutorial content only when it matches the current observation (layout, UI elements, stated goal). If the retrieved document appears irrelevant or mismatched, the model is guided to ignore it and rely on its own internal reasoning.
> > >
> > > Moreover, we also investigate the failure cases of R-WoM and here are the statistics of failure in terms of task types. Specifically, we follow WMA introduced in our paper to categorize tasks into seeking information or doing navigation and modifying content of a website/file/software.
> > >
> > > | Model             | Information-Seeking / Navigation | Content-Modification |
> > > |-------------------|----------------------------------|-----------------------|
> > > | Qwen2.5-VL-72B-Instruct    | 23.5%                            | 76.5%                 |
> > > | Claude-3.5-Sonnet        | 21.8%                            | 78.2%                 |
> > > | Claude-3.7-Sonnet        | 19.6%                            | 80.4%                 |
> > >
> > > | Model             | Information-Seeking / Navigation | Content-Modification |
> > > |-------------------|----------------------------------|-----------------------|
> > > | Qwen2.5-VL-72B-Instruct    | 37.4%                            | 62.6%                 |
> > > | Claude-3.5-Sonnet        | 35.7%                            | 64.3%                 |
> > > | Claude-3.7-Sonnet        | 34.8%                            | 65.2%                 |
> > >
> > > From the results, we observe that content-modification tasks (e.g., modify content in libreoffice/vscode) are where R-WoM fail more frequently. These failures often stem from ambiguous task goals (e.g., "Please calculate the ages of the employees according to their birthday"), where the model must infer missing steps, or from the inherent difficulty of manipulating fine-grained content through UI actions such as dragging, selecting, or highlighting text. Such tasks remain challenging for current computer-use agents because they require both precise visual grounding and multi-step reasoning. Exploring richer strategies, such as multi-modal retrieval or more agentic, step-aware retrieval mechanisms, can represent promising directions for future improvement.

---

> > > > ### Author Response · Authors · 2025-11-22
> > > > **Responses to Reviewer Cuhc (Part 4/4)**
> > > >
> > > > > ***Q2: Reranker details: Is it zero-shot or fine-tuned? How sensitive are results to the reranker model choice?***
> > > >
> > > > In our setting, the reranker is used in a zero-shot manner: we simply reuse the policy model itself as the reranker without any additional fine-tuning. To examine whether the choice of reranker model affects retrieval quality, we compare three policy models (Qwen-2.5-VL-72B-Instruct, Claude-3.5-Sonnet, and Claude-3.7-Sonnet) as rerankers on both OSWorld and WebArena. The results show only mild variations across models, with more capable models yielding slightly higher retrieval performance.
> > > >
> > > > | Reranker Model    | Dataset  | Recall@1 | Recall@3 | Recall@5 |
> > > > | ----------------- | -------- | -------- | -------- | -------- |
> > > > | Qwen-2.5-VL-72B-Instruct   | OSWorld  | 77.8     | 90.5     | 94.4     |
> > > > | Claude-3.5-Sonnet | OSWorld  | 79.5     | 92.0     | 95.2     |
> > > > | Claude-3.7-Sonnet | OSWorld  | 80.3     | 92.7     | 95.8     |
> > > > | Qwen-2.5-VL-72B-Instruct   | WebArena | 49.0     | 79.6     | 85.7     |
> > > > | Claude-3.5-Sonnet | WebArena | 50.8     | 81.1     | 87.0     |
> > > > | Claude-3.7-Sonnet | WebArena | 52.0     | 82.0     | 88.1     |
> > > >
> > > >
> > > > > ***Q3: Listwise reward computation: provide the exact mathematical definition (or pseudocode) of in Eq. (11). How are rewards normalized across rollouts, and how does candidate set size m affect ranking quality?***
> > > >
> > > > To compute the listwise reward, we follow a prompting-based formulation similar to recent works [7,8,9] that implement ranking rewards directly through LLM scoring. Instead of computing pairwise preferences, the judge LLM is asked to holistically compare the entire candidate action set and return a ranked list. This framing naturally aligns with the goal of selecting the globally best next action and avoids inconsistencies that may arise when aggregating multiple pairwise decisions. Because the reward is produced as a normalized ranking over the full list, it is inherently stable across different rollouts.
> > > >
> > > > To understand the effect of the candidate set size (m), we vary (m = 2, 3, 5) and measure downstream OSWorld performance. The results are shown below:
> > > >
> > > > | Action Candidate Size | Claude-3.7-Sonnet | Claude-4-Sonnet | Claude-4.5-Sonnet |
> > > > |-----------------------|------------|----------|-------------|
> > > > | 2                     | 37.90%     | 53.67%   | 66.00%      |
> > > > | 3                     | 39.13%     | 56.73%   | 67.84%      |
> > > > | 5                     | 38.10%     | 56.22%   | 68.41%      |
> > > >
> > > > We observe different behaviors across models when increasing the number of action candidates. For Claude-3.7-Sonnet and Claude-4-Sonnet, generating more actions at once often introduces redundant or low-quality candidates, which reduces the benefit of larger candidate sets. In contrast, Claude-4.5-Sonnet exhibits stronger action diversity and maintains higher action quality even when producing more candidates.
> > > >
> > > > [1] Yao et al. ReAct: Synergizing Reasoning and Acting in Language Models. ICLR 2023
> > > >
> > > > [2] Wu et al. AutoGen: Enabling Next-Gen LLM Applications via Multi-Agent Conversation. COLM 2024
> > > >
> > > > [3] He et al. WebVoyager: Building an End-to-End Web Agent with Large Multimodal Models. ACL 2024
> > > >
> > > > [4] Wang et al. Agent Workflow Memory. ICML 2025
> > > >
> > > > [5] Wang et al. Inducing Programmatic Skills for Agentic Tasks. COLM 2025
> > > >
> > > > [6] Wang et al. OPENCUA: Open Foundations for Computer-Use Agents. NeurIPS 2025
> > > >
> > > > [7] Sun et al. Is ChatGPT Good at Search? Investigating Large Language Models as Re-Ranking Agents. EMNLP 2023
> > > >
> > > > [8] Qin et al. Large Language Models are Effective Text Rankers with Pairwise Ranking Prompting. NAACL 2024 Findings
> > > >
> > > > [9] Yu et al. RankRAG: Unifying Context Ranking with Retrieval-Augmented Generation in LLMs. NeurIPS 2024

---

### Official Review · Reviewer_W8D9 · 2025-11-03

**Soundness:** 2
**Presentation:** 3
**Contribution:** 2
**Rating:** 4
**Confidence:** 4

**Summary:**

This paper addresses an important challenge in developing computer-use agents: enabling LLMs to serve as effective world models for long-horizon planning in digital environments. The authors begin with a thoughtful diagnostic study using three probing tasks that reveal LLMs excel at short-term state predictions (86% accuracy) but struggle with long-horizon planning alignment (50-65%). To address this limitation, they propose R-WoM (Retrieval-augmented World Model), which grounds LLM-based simulations with external tutorials retrieved at inference time. The framework employs chain-of-thought rollouts and listwise reward estimation for action selection. Experiments on WebArena and OSWorld demonstrate improvements of up to 25.3% and 18.1% respectively over the WebDreamer baseline.

**Strengths:**

**1. Well-Motivated Diagnostic Study:** I appreciate the systematic probing analysis in Section 3. The three-task evaluation (next-state identification, full-procedure planning alignment, milestone transition recognition) provides clear empirical evidence for why existing LLM-based world models struggle with long-horizon planning. This diagnostic approach effectively motivates the need for external grounding mechanisms.

**2. Reasonable Architectural Integration:** The paper successfully brings together two established paradigms—world models from model-based RL and retrieval-augmented generation—and applies them to the challenging computer-use domain. I find this combination sensible and well-suited to the problem.

**3. Comprehensive Ablation Studies:** I commend the authors for including ablations on imagination horizons (Table 3) and grounding sources (retrieved vs. oracle tutorials in Figure 3). These studies help understand which components contribute to the overall performance.

**4. Practical Efficiency Considerations:** The use of CoT-based rollouts instead of expensive iterative interactions represents a pragmatic design choice. Table 5 shows R-WoM is more efficient than WebDreamer, which is valuable for practical deployment.

**5. Solid Engineering Effort:** The collection of 30k+ tutorial chunks from diverse sources and the design of the reasoning-based RAG pipeline with query rewriting and reranking demonstrates significant implementation effort.

**Weaknesses:**

**1. Limited Baseline Comparisons:** I am concerned that the experimental evaluation could be more comprehensive. The paper primarily compares against WebDreamer and basic policy models, but I understand there are other benchmarks in this space that could provide important context. For example, I noticed that OpenAI's Computer Use Agent (CUA) published results showing 58.1% on WebArena (available since January 2025), which would be a valuable reference point. I recognize the authors' results of 35.11% show improvement over their chosen baseline, but including broader comparisons would help readers better understand where R-WoM stands in the current landscape. I encourage the authors to discuss or compare with additional publicly available benchmarks to strengthen the evaluation.

**2. Characterization of Related Methods:** I noticed that the paper characterizes some related work in ways that may not fully capture their methodologies. For instance, lines 310-316 discuss WebEvolver (Fang et al., 2025) in the context of "absolute reward estimation," but from my reading, WebEvolver appears to use supervised fine-tuning with rejection sampling rather than RL-based absolute rewards. I suggest the authors revisit these characterizations to ensure they accurately represent the cited methods, which would strengthen the positioning of the listwise reward contribution.

**3. Retrieval Quality Analysis:** While Figure 3 shows that oracle tutorials outperform retrieved ones, I would appreciate more analysis of how retrieval failures affect performance. The appendix mentions Recall@5 metrics (94.4% OSWorld, 85.7% WebArena), suggesting some retrieval errors occur. I recommend adding analysis of what types of tasks or scenarios are most affected by retrieval failures and how these errors propagate through the world model rollouts.

**4. Computational Cost Discussion:** Table 5 shows R-WoM requires 3.8-12h runtime compared to 0.7-2.1h for vanilla methods—a 4-10× increase. While this is better than WebDreamer's 15-43h, I believe the practical deployment implications deserve more discussion in the main paper. For real-world applications, this trade-off between performance and computational cost would be important to consider.

**5. Gap Between Diagnosis and Solution:** The probing study identifies 50-65% full-procedure planning alignment as the core problem. However, I would be interested to see whether R-WoM actually improves this alignment metric, or if the task success improvements come from other mechanisms. This would help validate that the solution addresses the diagnosed problem.

**Questions:**

**Q1:** Could you provide more context on how your results compare with other available benchmarks in this space? Specifically, I noticed OpenAI CUA reported 58.1% on WebArena in early 2025. Are there methodological or setup differences that make direct comparison inappropriate, or would it be valuable to discuss these results to help readers understand the current state of the field?

**Q2:** Regarding the characterization of WebEvolver using "absolute reward estimation" (line 311): could you clarify this? From my understanding, WebEvolver uses supervised fine-tuning with rejection sampling. I would appreciate clarification on how you're categorizing their reward mechanism.

**Q3:** Your probing study identifies long-horizon planning alignment as the key limitation (50-65% accuracy). After applying R-WoM with tutorial grounding, what is the new alignment score on this metric? This would help demonstrate that your solution directly addresses the diagnosed problem.

**Q4:** Could you provide more analysis of retrieval failure cases? What types of tasks or scenarios are most affected when the retrieval system fails to find appropriate tutorials, and how do these failures impact the final task success?

**Q5:** Have you considered any techniques to reduce the 4-10× computational overhead? Are there scenarios where the performance gains justify this cost, and others where a lighter-weight approach might be preferable?

---

> ### Author Response · Authors · 2025-11-22
> **Responses to Reviewer W8D9 (Part 1/3)**
>
> > ***W1: Limited Baseline Comparisons: I am concerned that the experimental evaluation could be more comprehensive. The paper primarily compares against WebDreamer and basic policy models, but I understand there are other benchmarks in this space that could provide important context. For example, I noticed that OpenAI's Computer Use Agent (CUA) published results showing 58.1% on WebArena (available since January 2025), which would be a valuable reference point. I recognize the authors' results of 35.11% show improvement over their chosen baseline, but including broader comparisons would help readers better understand where R-WoM stands in the current landscape. I encourage the authors to discuss or compare with additional publicly available benchmarks to strengthen the evaluation.***
>
> > ***Q1: Could you provide more context on how your results compare with other available benchmarks in this space? Specifically, I noticed OpenAI CUA reported 58.1% on WebArena in early 2025. Are there methodological or setup differences that make direct comparison inappropriate, or would it be valuable to discuss these results to help readers understand the current state of the field?***
>
> Please refer to our response in the general response.
>
> > ***W2: Characterization of Related Methods: I noticed that the paper characterizes some related work in ways that may not fully capture their methodologies. For instance, lines 310-316 discuss WebEvolver (Fang et al., 2025) in the context of "absolute reward estimation," but from my reading, WebEvolver appears to use supervised fine-tuning with rejection sampling rather than RL-based absolute rewards. I suggest the authors revisit these characterizations to ensure they accurately represent the cited methods, which would strengthen the positioning of the listwise reward contribution.***
>
> > ***Q2: Regarding the characterization of WebEvolver using "absolute reward estimation" (line 311): could you clarify this? From my understanding, WebEvolver uses supervised fine-tuning with rejection sampling. I would appreciate clarification on how you're categorizing their reward mechanism.***
>
> We appreciate your careful check of our related works (i.e., WebEvolver). WebEvolver uses the scalar score reward during inference time, which is the core comparison point for our listwise reward design, as stated in the their original paper:
>
> > "Following Koh et al. (2024b); Gu et al. (2024), the evaluator assigns a scalar from {0, 0.5, 1.0} (incorrect, on track, or complete) based on the trajectory’s alignment with task completion."
>
> This inference-time reward assignment contrasts with our proposed listwise approach.

---

> ### Author Response · Authors · 2025-11-22
> **Responses to Reviewer W8D9 (Part 2/3)**
>
> > ***W3: Retrieval Quality Analysis: While Figure 3 shows that oracle tutorials outperform retrieved ones, I would appreciate more analysis of how retrieval failures affect performance. The appendix mentions Recall@5 metrics (94.4% OSWorld, 85.7% WebArena), suggesting some retrieval errors occur. I recommend adding analysis of what types of tasks or scenarios are most affected by retrieval failures and how these errors propagate through the world model rollouts.***
>
> > ***Q4: Could you provide more analysis of retrieval failure cases? What types of tasks or scenarios are most affected when the retrieval system fails to find appropriate tutorials, and how do these failures impact the final task success?***
>
> Thank you for your suggestions. We have conducted a detailed analysis of task achievement related to retrieval failure in R-WoM. The tables below summarize these statistics:
>
> **Statistics on OSWorld**
>
> | Metric                                                   | Qwen2.5-VL-72B-Instruct | Claude-3.5-Sonnet | Claude-3.7-Sonnet |
> | -------------------------------------------------------- | -------------- | ----------------- | ----------------- |
> | Retrieval failures                                       | 5              | 4                 | 4                 |
> | Successful tasks despite retrieval failure               | 3              | 3                 | 3                 |
> | Failures shared with WebDreamer (i.e., not caused by retrieval) | 2              | 1                 | 1                 |
>
> **Statistics on WebArena**
>
> | Metric                                                   | Qwen2.5-VL-72B-Instruct | Claude-3.5-Sonnet | Claude-3.7-Sonnet |
> | -------------------------------------------------------- | -------------- | ----------------- | ----------------- |
> | Retrieval failures                                       | 17             | 17                | 16                |
> | Successful tasks despite retrieval failure               | 11             | 12                | 11                |
> | Failures shared with WebDreamer (i.e., not caused by retrieval) | 6              | 5                 | 5                 |
>
> These results show that even when retrieval returns suboptimal content, the agent can still complete some tasks, and the remaining failures overlap heavily with the baseline WoM rather than being caused by irrelevant tutorials. This result suggests that the retrieval failure in R-WoM do not bring much additional misguidance on the agents, indicating the robustness of our framework.
>
> Moreover, we also investigate the failure cases of R-WoM and here are the statistics of failure in terms of task types. Specifically, we follow the WMA [1] to categorize tasks into 1) seeking information or doing navigation and 2) modifying content of a website/file/software.
>
> **On OSWorld**
>
> | Model             | Information-Seeking / Navigation | Content-Modification |
> |-------------------|----------------------------------|-----------------------|
> | Qwen-2.5-VL-72B-Instruct    | 23.5%                            | 76.5%                 |
> | Claude-3.5-Sonnet        | 21.8%                            | 78.2%                 |
> | Claude-3.7-Sonnet        | 19.6%                            | 80.4%                 |
>
> **On WebArena**
>
> | Model             | Information-Seeking / Navigation | Content-Modification |
> |-------------------|----------------------------------|-----------------------|
> | Qwen2.5-VL-72B-Instruct    | 37.4%                            | 62.6%                 |
> | Claude-3.5-Sonnet        | 35.7%                            | 64.3%                 |
> | Claude-3.7-Sonnet        | 34.8%                            | 65.2%                 |
>
> From the results, we observe that content-modification tasks (e.g., modify content in libreoffice/vscode) are where R-WoM fails more frequently. These failures often stem from ambiguous task goals (e.g., “Please calculate the ages of the employees according to their birthday”), where the model must infer missing steps, or from the inherent difficulty of manipulating fine-grained content through UI actions such as dragging, selecting, or highlighting text. Such tasks remain challenging for current computer-use agents because they require both precise visual grounding and multi-step reasoning. Exploring richer strategies, such as multi-modal retrieval or more agentic, step-aware retrieval mechanisms, represents a promising direction for future improvement.

---

> ### Author Response · Authors · 2025-11-22
> **Responses to Reviewer W8D9 (Part 3/3)**
>
> > ***W4: Computational Cost Discussion: Table 5 shows R-WoM requires 3.8-12h runtime compared to 0.7-2.1h for vanilla methods—a 4-10× increase. While this is better than WebDreamer's 15-43h, I believe the practical deployment implications deserve more discussion in the main paper. For real-world applications, this trade-off between performance and computational cost would be important to consider.***
>
> > ***Q5: Have you considered any techniques to reduce the 4-10× computational overhead? Are there scenarios where the performance gains justify this cost, and others where a lighter-weight approach might be preferable?***
>
> Please refer to our response in the general response.
>
> > ***W5: Gap Between Diagnosis and Solution: The probing study identifies 50-65% full-procedure planning alignment as the core problem. However, I would be interested to see whether R-WoM actually improves this alignment metric, or if the task success improvements come from other mechanisms. This would help validate that the solution addresses the diagnosed problem.***
>
> > ***Q3: Your probing study identifies long-horizon planning alignment as the key limitation (50-65% accuracy). After applying R-WoM with tutorial grounding, what is the new alignment score on this metric? This would help demonstrate that your solution directly addresses the diagnosed problem.***
>
> Thank you for bringing this insightful suggestion. After applying retrieval-augmentation, we directly re-measured the long-horizon planning alignment accuracy reported in our probing study. The updated results are shown below:
>
> | Model              | Planning Alignment (w/o Retrieval) | Planning Alignment (w/ Retrieval) |
> |--------------------|-------------------------------------|------------------------------------|
> | Qwen-2.5-VL-72B-Instruct     | 50.0%                               | 90.0%                              |
> | Claude-3.5-Sonnet   | 55.0%                               | 85.0%                              |
> | Claude-3.7-Sonnet   | 65.0%                               | 95.0%                              |
>
> The results show that using retrieval-augmentation, models can generate more reliable plans over long-horizons, which aligns with our results in the end-to-end performance.
>
> [1] Chae et al. Web Agents with World Models: Learning and Leveraging Environment Dynamics in Web Navigation. ICLR 2025

---

### Author Response · Authors · 2025-11-22
**General Response (Part 1/2)**

We thank all the reviewers for their valuable suggestions and feedbacks. We provide more clarifications and additional experimental results, which we hope can address reviewers' concerns.

The following sections are our responses to issues that appeared across multiple reviews.

> ***Comparison with more baselines***

We would like to highlight that our R-WoM is a plug-and-play component that can be applied to different models. This design makes our approach broadly compatible and model-agnostic. To further address the reviewers' concern regarding comparison with stronger state-of-the-art baselines, we additionally evaluated our method using more powerful recent LLMs (Claude-4-Sonnet and Claude-4.5-Sonnet), both of which have been optimized for computer-use environments.

We evaluate them on the OSWorld benchmark using the same setting in the Section 5.1 of our paper and the result is shown as below.

| Model    | Claude-4-Sonnet | Claude-4.5-Sonnet |
|----------|----------|-------------|
| Vanilla  | 48.24%   | 59.12%      |
| RAG      | 50.82%   | 60.43%      |
| WoM      | 49.65%   | 62.09%      |
| **R-WoM** | **56.73%** | **67.84%** |

On these stronger models, we can also observe consistent performance improvements when applying our method, demonstrating its effectiveness. More discussions of comparison with state-of-the-art baselines will be included in our revised paper.

---

> ### Author Response · Authors · 2025-11-22
> **General Response (Part 2/2)**
>
> > ***Deeper Discussion of Cost-Performance Tradeoff***
>
> Similar to other test-time scaling approaches, the cost of R-WoM inevitably increases because some intermediate rollouts or expanded action candidates are not always necessary. However, test-time scaling remains valuable because it enables stronger reasoning without additional training, and R-WoM is already more efficient than other world model methods, e.g. WebDreamer, by eliminating communication between the policy and the world model during rollout.
>
> To further improve cost efficiency, we introduce an adaptive version of R-WoM. The adaptive design is motivated by the observation that not every step of a trajectory requires world-model reasoning. It incorporates two mechanisms:
>
> 1. Adaptive action branching. The policy model adaptively decides when to generate more or fewer action candidates, preventing redundant action generation. In particular, if the policy model only generates a single action, there is no need to trigger the world model.
> 2. Action deduplication. A verifier removes duplicated action candidates, avoiding unnecessary branches that would otherwise trigger redundant world-model rollouts.
>
> We apply these cost-optimization strategies on Claude (3.7-Sonnet, 4-Sonnet and 4.5-Sonnet) to run on OSWorld and evaluate both performance and token usage as below:
>
> | Model            | Claude-3.7-Sonnet | Claude-4-Sonnet | Claude-4.5-Sonnet |
> | ---------------- | ---------- | -------- | ---------- |
> | Vanilla          | 29.40%     | 48.24%   | 59.12%     |
> | RAG              | 27.76%     | 50.82%   | 60.43%     |
> | WebDreamer       | 31.24%     | 49.65%   | 62.09%     |
> | R-WoM            | 39.13%     | 56.73%   | 67.84%     |
> | R-WoM (Adaptive) | 37.80%     | 55.18%   | 66.37%     |
>
> Above results on performance show that adaptive R-WoM preserves nearly all of R-WoM advantages: for Claude-4.5-Sonnet, performance rises from 59.12% (Vanilla) and 60.43% (RAG) to 66.37% with R-WoM (Adaptive), a relative gain of about 12% over Vanilla and 10% over RAG while staying very close to the full R-WoM score of 67.84%.
>
> We measure how much the adaptive mechanisms reduce unnecessary action branches, duplicated actions after generation, and excessive world-model rollouts.
>
> | Model      | Adaptive Action branching | Action Deduplication | Final World-Model Trigger Reduction |
> | ---------- | -------------------- | --------------------------- | ------------------------ |
> | Claude-3.7-Sonnet | 30.8%                | 31.5%                       | 80.5%                    |
> | Claude-4-Sonnet   | 29.7%                | 34.2%                       | 82.2%                    |
> | Claude-4.5-Sonnet | 28.4%                | 38.1%                       | 84.4%                    |
>
> From the above table, the adaptive version reduces multi-action generation to roughly 30% of the original and removes 30–40% of duplicated triggers. Accordingly, the world-model triggering rate drops to 15–20%.
>
> We then evaluate the token usage across different methods to see whether the adaptive R-WoM is indeed cost-efficient.
>
> | Model            | Claude-3.7-Sonnet Input | Claude-3.7-Sonnet Output | Claude-4-Sonnet Input | Claude-4-Sonnet Output | Claude-4.5-Sonnet Input | Claude-4.5-Sonnet Output |
> | ---------------- | ---------------- | ----------------- | -------------- | --------------- | ---------------- | ----------------- |
> | Vanilla          | 32.45M           | 0.62M             | 29.91M         | 0.37M           | 29.35M           | 0.36M             |
> | RAG              | 34.39M           | 0.66M             | 31.69M         | 0.38M           | 31.10M           | 0.38M             |
> | WebDreamer       | 226.37M          | 4.35M             | 208.60M        | 2.56M           | 204.69M          | 2.51M             |
> | R-WoM            | 76.19M           | 1.47M             | 70.27M         | 0.86M           | 68.95M           | 0.85M             |
> | R-WoM (Adaptive) | 35.39M           | 0.85M             | 32.60M         | 0.50M           | 31.94M           | 0.49M             |
>
> Compared to full R-WoM, the adaptive variant reduces token usage by more than 50% in most settings. Importantly, its total cost becomes much closer to RAG: for instance, on Claude-4.5-Sonnet, R-WoM adaptive uses only slightly more input and output tokens than RAG (within roughly 5–10%). The results suggest that our adaptive design can help maintain most of the performance gain in R-WoM while substantially reducing token consumption.

---

### Author Response · Authors · 2025-12-02

Dear Reviewers, ACs, SACs, and PCs,

We would like to summarize the key strengths of our work as acknowledged by the reviewers, and to outline the additional responses and revisions we have made to address all major concerns.

Strengths acknowledged by the reviewers

1. Strong Motivation (**Reviewer W8D9**, **Reviewer Cuhc**, **Reviewer tkQf**):
   Our work is well motivated by a systematic diagnosis of current LLM world-modeling limitations in multi-turn realistic environments, with the goal of improving long-horizon computer-use capabilities of LLM-based agents.

2. Practical Framework Design (**Reviewer W8D9**, **Reviewer Cuhc**, **Reviewer tkQf**, **Reviewer 1XKm**):
   The proposed framework is model- and agent-agnostic and can be seamlessly integrated into existing LLM agent stacks.

3. Extensive Evaluation and Actionable Insights (**Reviewer W8D9**, **Reviewer Cuhc**, **Reviewer tkQf**, **Reviewer 1XKm**):
   Our experiments demonstrate consistent, and in some cases substantial, performance gains across multiple models and two challenging benchmarks. The ablation studies on imagination horizons and grounding sources (retrieved vs. oracle tutorials, Figure 3) further provide actionable insights for the community.

To further strengthen the paper, we conducted additional experiments and clarifications to address the following key concerns:

1. Cost–Performance Trade-off and Further Cost Optimization (**Reviewer W8D9**, **Reviewer tkQf**)
2. Extension to Tutorial-Scarce Domains (**Reviewer Cuhc**, **Reviewer tkQf**)
3. More Comprehensive Failure Case Analysis (**Reviewer W8D9**, **Reviewer Cuhc**, **Reviewer tkQf**, **Reviewer 1XKm**)

The additional results and revisions are also updated in our paper. We hope that the additional results and revisions made during the rebuttal phase can be carefully considered.

Thank you very much for your time and thoughtful feedback.

Sincerely,
R-WoM Authors

---

### Meta-Review · Area_Chair_F1TZ · 2026-01-07

**Summary:**

This paper addresses a central challenge in building computer-use agents: enabling large language models to function as effective world models for long-horizon planning in digital environments. The authors begin with a careful diagnostic analysis using three probing tasks, showing that while LLMs achieve strong performance on short-term state prediction (up to 86% accuracy), they exhibit substantial deficiencies in long-horizon planning alignment (50–65%).

To mitigate this gap, the paper proposes R-WoM (Retrieval-augmented World Model), a framework that grounds LLM-based simulation through inference-time retrieval of external tutorials. The method integrates chain-of-thought rollouts with listwise reward estimation to guide action selection. Empirical results on WebArena and OSWorld demonstrate consistent and meaningful gains, achieving improvements of up to 25.3% and 18.1%, respectively, over the WebDreamer baseline. Overall, the work presents a diagnostic-driven and empirically validated approach that contributes practical insights into improving long-horizon planning for computer-use agents.

**Reviewer Concerns:**

Reviewers generally agree that the work is technically sound and carefully executed, with solid engineering effort and thorough experimentation. At the same time, concerns are raised about the limited conceptual novelty, as the framework builds on established components (RAG, world models, CoT), and about practical trade-offs, including increased computational cost and reliance on tutorial availability. Some reviewers also question evaluation breadth and generalization to tutorial-scarce or highly novel environments.

The authors’ rebuttal substantially addresses several of these concerns. In particular, they provide new evidence that retrieval augmentation directly improves the originally diagnosed planning-alignment metric, clarify related-work positioning, analyze retrieval failures and failure modes in depth, and demonstrate that R-WoM remains effective even in tutorial-scarce settings via synthesized procedural knowledge. They also contextualize the computational overhead and show that retrieval itself contributes only a small fraction of total runtime.

**Reviewer Scores:**

Reviewer W8D9.
Initially rated the paper slightly below the acceptance threshold, with primary concerns about whether the proposed method meaningfully addressed the diagnosed long-horizon planning misalignment. After discussion and the authors’ additional analysis demonstrating clear improvements on the planning-alignment metric, these concerns appear largely resolved. The reviewer would likely revise their score upward to slightly above the acceptance threshold.

Reviewer Cuhc.
Provided an above-threshold score in the initial review, noting strong empirical results but expressing reservations about conceptual novelty and practical trade-offs. These concerns were partially alleviated during discussion through clarified positioning and additional ablations. The reviewer would likely maintain an above-threshold score, with confidence in the paper’s empirical contribution.

Reviewer 1XKm.
Gave an above-threshold score and was generally positive about the methodology and experimental thoroughness, while raising questions about scalability and broader generalization. The discussion did not introduce new issues that would materially change this assessment. The reviewer would likely keep their original score.

Reviewer tkQf.
Initially scored the paper below threshold, citing concerns about realism, reliance on external tutorials, and limited conceptual novelty. While the authors’ rebuttal addressed several points through new experiments and clarifications, some reservations remain. The reviewer would likely increase their score slightly but still remain below the acceptance threshold.

---

### Decision · Program_Chairs · 2026-01-26

Accept (Poster)